# Predictable Scale (Part II)
# — Farseer: A Refined Scaling Law in LLMs

**Houyi Li**[*]
Fudan University & StepFun, China

**Wenzhen Zheng**[*]
StepFun, China

**Qiufeng Wang**
StepFun, China

**Zhenyu Ding**
Xi'an Jiaotong University, China

**Haoying Wang**
Xi'an Jiaotong University, China

**Zili Wang**
StepFun, China

**Shijie Xuyang**
Fudan University & StepFun, China

**Ning Ding**[†]
Xi'an Jiaotong University, China

**Shuigeng Zhou**[†]
Fudan University, China

**Xiangyu Zhang**
StepFun & Megvii Technology, China

**Daxin Jiang**
StepFun, China

## Abstract

Training Large Language Models (LLMs) is prohibitively expensive, creating a critical *scaling gap* where insights from small-scale experiments often fail to transfer to resource-intensive production systems, thereby hindering efficient innovation. To bridge this, we introduce **Farseer**, a novel and refined scaling law offering enhanced predictive accuracy across scales. By systematically constructing a model loss surface $L(N, D)$, *Farseer* achieves a significantly better fit to empirical data than prior laws (e.g., *Chinchilla's law*). Our methodology yields accurate, robust, and highly generalizable predictions, demonstrating excellent extrapolation capabilities, outperforming Chinchilla's law, whose extrapolation error is 433% higher. This allows for the reliable evaluation of competing training strategies across all $(N, D)$ settings, enabling conclusions from small-scale ablation studies to be confidently extrapolated to predict large-scale performance. Furthermore, *Farseer* provides new insights into optimal compute allocation, better reflecting the nuanced demands of modern LLM training. To validate our approach, we trained an extensive suite of approximately 1,000 LLMs across diverse scales and configurations, consuming roughly 3 million NVIDIA H100 GPU hours. To foster further research, we are comprehensively open-sourcing all code, data, results [3], all training logs[4], all models used in scaling law fitting [5].

---

[*]Equal contribution.

[†]Corresponding author. Ning Ding: ding.ning@xjtu.edu.cn; Shuigeng Zhou: sgzhou@fudan.edu.cn

[3]https://github.com/Farseer-Scaling-Law/Farseer

[4]https://wandb.ai/billzid/Farseer?nw=nwuserbillzid

[5]https://huggingface.co/Farseer-Scaling-Law

39th Conference on Neural Information Processing Systems (NeurIPS 2025).

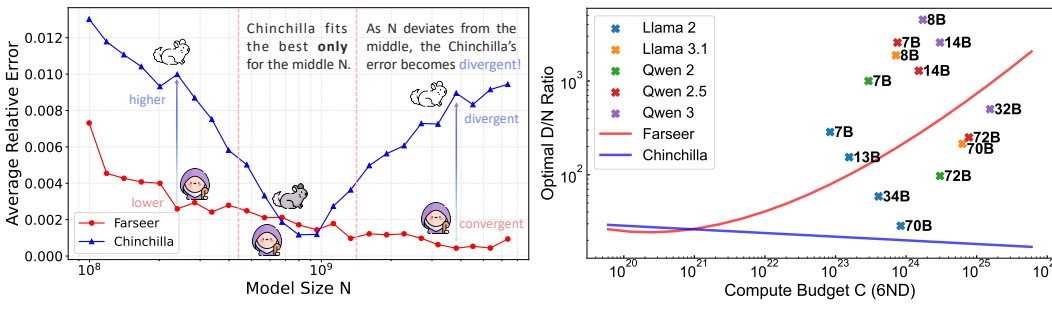

(a) Comparison of Average Relative Error  (b) Optimal $D/N$ Ratio vs. Compute Budget

Figure 1: ***Farseer* beats Chichilla** [20]. (a) Average relative error (BPC) vs. model size $N$ for *Farseer* (red) and Chinchilla (blue). Chinchilla, lacking high-order cross terms, fits only near the central $N$ and its error diverges as model size grows. In contrast, ***Farseer*'s error is 232% lower within the fitted range and remains stable across the full $N$ range.** (b) Chinchilla's rule of thumb ($D/N \approx 20$) is valid only at moderate budgets ($C \approx 10^{20} - 10^{21}$), but it underestimates the requirements for larger scale regimes. In contrast, our analysis predicts a steadily increasing optimal $D/N$, which is consistent with the actual training configurations used in recent large language models (e.g., Llama 3.1 [16], Qwen3 [45], etc.).

## 1 Introduction

Recent remarkable progress in Large Language Models (LLMs) such as GPT-3 [6], GPT-4 [2], and Llama [39] is largely attributed to scaling laws, notably proposed by Kaplan et al. [22]. These laws demonstrate that model performance, typically measured by loss $L$, exhibits a predictable improvement trend as model parameters ($N$) and training data ($D$) increase. This relationship follows power-law dynamics expressed as:

$$L(N, D) = \left[ \left( \frac{N_c}{N} \right)^{\frac{\alpha_N}{\alpha_D}} + \frac{D_c}{D} \right]^{\alpha_D}, \tag{1}$$

Subsequently, DeepMind's Chinchilla [20] proposed optimal compute-scaling strategies and a revised scaling law:

$$L(N, D) \approx \frac{A}{N^\alpha} + \frac{B}{D^\beta} + E. \tag{2}$$

where all parameters other than $N$ and $D$ are fitted. While valuable, Chinchilla's formulation has limitations in modeling the interplay between model size and data scaling. Specifically, we argue that its term $\frac{B}{D^\beta}$, which describes how loss improves with data $D$, uses constant parameters $B$ and $\beta$. This implies that a rate of improvement with data is uniform across all model sizes $N$, thereby lacking adequate modeling of $N$'s influence on data scaling dynamics. Consequently, Chinchilla's law tends to capture an average data scaling behavior across the $N$ values used for fitting. As a result, it performs best for models near the midpoint of this range, but less accurately at the extremes of $N$, as

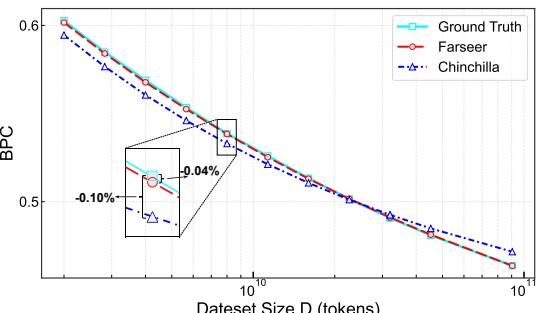

Figure 2: Empirical BPC values (Ground Truth) are plotted alongside fits from *Farseer* and Chinchilla for a fixed model size of $N = 6B$. *Farseer* yields predictions that lie almost exactly on the ground truth curve, whereas Chinchilla's fit exhibits systematic under- and over-estimations, particularly at small and large $D$.

shown in Fig. 1 (a). This characteristic inherently limits its extrapolation capabilities, especially for model sizes significantly different from its calibration set, making cost-effective prediction across arbitrary $(N, D)$ surfaces a persistent challenge.

These limitations in accurately predicting performance with existing scaling laws underscore a broader difficulty: the very exploration of superior scaling laws is severely hampered by a significant *scaling gap*. The immense computational cost of state-of-the-art LLM training (often $> 10^{25}$ FLOPs) means insights from affordable, small-scale experiments frequently fail to transfer to production scales. This "scaling gap" refers to the phenomenon where conclusions drawn from small-scale experiments do not consistently hold when scaling up to larger models. A more accurate scaling law can mitigate this by providing more reliable predictions, making small-scale exploratory experiments more valuable (see Appendix E for an example).

To bridge this gap, we introduce *Farseer*, a refined scaling law (Eq. 3) and an experimental methodology developed from training over 1,000 LLMs. *Farseer* employs **Differential Piecewise** and **Multi-round Iterative fitting** to model the loss surface $L(N, D)$:

$$L(N, D) = e^{a_3 \cdot N^{\gamma} + b_3} + e^{a_2 \cdot N^{\beta} + b_2} \cdot D^{-e^{a_1 \cdot N^{\alpha} + b_1}} \tag{3}$$

where all parameters other than $N$ and $D$ are fitted. Our key contributions are:

- **Refined scaling law.** We propose *Farseer* (Eq. 3), providing a significantly more accurate fit to empirical LLM data (Fig. 1 (a) and 2) through a novel fitting approach where data scaling effects are explicitly $N$-dependent.
- **Superior extrapolation.** *Farseer* enables reliable large-scale performance prediction from tractable small-scale experiments, effectively bridging the scaling gap.
- **Improved compute guidance.** Our analysis yields new, data-driven insights for optimal $D/N$ allocation in modern LLM training, diverging from simpler heuristics (Fig. 1 (b)).
- **Comprehensive open-sourcing.** We release all models, data from $\sim$1,000 trained LLMs, detailed logs, and the *Farseer* code to foster further research.

## 2 Preliminaries

### 2.1 General Loss Formulation

Our *Farseer* method systematically samples $L(N, D)$ via small-scale experiments (smaller $N, D$) and applies a mathematical scaling formulation to predict performance at significantly larger scales. This enables robust assessment of a training strategy's scaling potential.

For clarity, we decompose the loss $L(N, D)$ as:

$$L(N, D) = E + L_N(N) + L_D(D) + L_{ND}(N, D) \tag{4}$$

where $E$ is a constant term, $L_N(N)$ is the model-size-dependent loss component, $L_D(D)$ is the data-size-dependent loss component, and $L_{ND}(N, D)$ represents the interaction effect between $N$ and $D$. *Farseer* focuses on determining the functional forms and parameters of $L_N(N)$, $L_D(D)$ and $L_{ND}(N, D)$ using only small-scale experimental data.

### 2.2 Basic Settings

We train approximately 1,000 models with a standard language modeling objective [41, 11, 31]. The training data comprises a mix of web text, mathematical content, and code. This data is processed using a Byte Pair Encoding (BPE) [14] tokenizer with a vocabulary size of 65,536. We evaluated two distinct data mixtures following [39, 25], with specific component weightings detailed in Appendix B.

Our model architecture design follows the Llama [39, 16] design, using the AdamW optimizer [29] with $\beta$ values of [0.9, 0.95], an epsilon of $10^{-8}$, a weight decay of 0.1, and a gradient clipping norm of 1.0. We set the parameters $N$ and $D$ for these models using a geometric progression with a common ratio of $\sqrt{2}$, and specific details can be found in the Appendix F. A visualization of the experimental $(N, D)$ grid is provided in Fig. 7 (blue circles). Our learning rate schedule includes a linear warmup for the first 2,000 steps, followed by cosine decay to $1 \times 10^{-5}$ for the remainder of training. The model uses a fixed sequence length of 2,048 tokens. Ultimately, we define model size $N$ by excluding embedding layer parameters (see Appendix G for details). Further elaboration on the metric computations, and considerations for optimal hyperparameter settings [25], including model aspect ratios, can be found in Appendix A.

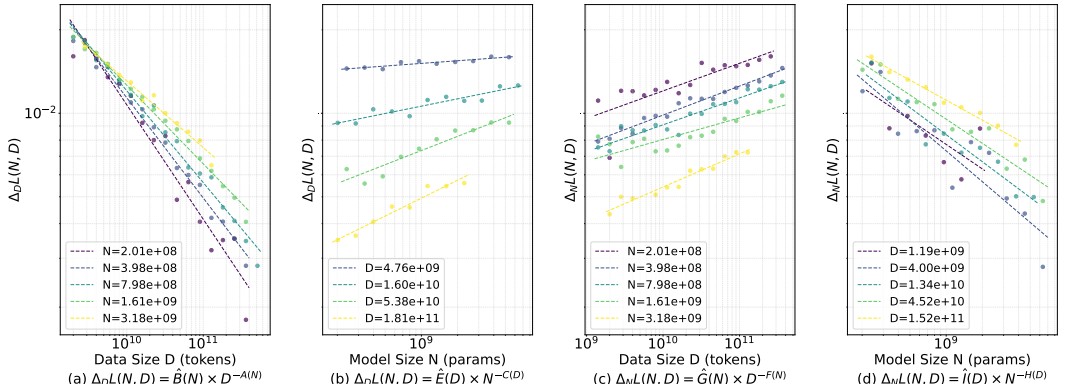

Figure 3: Log-Log analysis of the differential BPC terms $\Delta_D L$ and $\Delta_N L$ as univariate functions of $D$ and $N$, respectively: (a) $\Delta_D L$ vs. $D$ at fixed $N$ ($R^2 = 0.9807$); (b) $\Delta_D L$ vs. $N$ at fixed $D$ ($R^2 = 0.7457$); (c) $\Delta_N L$ vs. $D$ at fixed $N$ ($R^2 = 0.8484$); (d) $\Delta_N L$ vs. $N$ at fixed $D$ ($R^2 = 0.8733$). The high $R^2$ in (a) (0.9807) demonstrates a consistent power-law relationship between $\Delta_D L$ and $D$, so we adopt this form in our main analysis. The $R^2$ value is the average across all fitted lines. As detailed in Appendix I, the associated terms $A(N)$ and $\hat{B}(N)$ further offer improved numerical behavior.

For evaluation, we utilize a high-quality, specially constructed validation set containing 30 million tokens. This dataset is entirely separate from our training data, ensuring that all validation samples are unseen. It comprises a diverse mix of web pages, books, and academic papers, rigorously filtered to be more distilled and of higher quality than the training corpus. Instead of validation loss, we employ Bits Per Character (BPC) as our primary evaluation metric to measure the model's compression efficiency on this validation set. Further details are provided in Appendix D.

## 3 Methodology

This section summarizes our **Differential Piecewise Fitting** methodology for the performance surface $L(N, D) = L_D(D) + L_{ND}(N, D) + E + L_N(N)$. Section 3.1 presents a differential analysis establishing the necessity of the interaction term $L_{ND}(N, D)$. Section 3.2 details the modeling of the combined data-dependent terms $L_D(D) + L_{ND}(N, D)$ (Stages 1 & 2), guided by empirical observations to a power-law form in $D$. Appendix C provides the full algorithm. Finally, Section 3.3 describes fitting the model-dependent term $E + L_N(N)$ (Stage 3). The overall procedure yields a fitted scaling law: $L(N, D) \approx f_B(N; \theta_B^*) D^{-f_A(N; \theta_A^*)} + f_{L_N}(N; \theta_{L_N}^*)$.

### 3.1 Differential Analysis for the Existence of Higher-Order Interaction Term $L_{ND}(N, D)$

To understand the contributions of the terms in Eq. (4), we employ a differential technique. We compute finite differences of the loss $L(N, D)$:

- $\Delta_D L(N, D) = L(N, D) - L(N, \lambda D)$, this difference primarily reflects changes in the data-dependent terms: $\Delta_D L(N, D) \approx [L_D(D) - L_D(\lambda D)] + [L_{ND}(N, D) - L_{ND}(N, \lambda D)]$. This operation cancels $E$ and $L_N(N)$.

- $\Delta_N L(N, D) = L(N, D) - L(\lambda N, D)$. Similarly, this difference highlights changes in model-size dependent terms: $\Delta_N L(N, D) \approx [L_N(N) - L_N(\lambda N)] + [L_{ND}(N, D) - L_{ND}(N, \lambda D)]$. This cancels $E$ and $L_D(D)$.

Empirical analysis of these differential terms, as shown in Fig. 3, compellingly demonstrates the necessity of a non-degenerate interaction term $L_{ND}(N, D)$:

- $\Delta_D L(N, D)$ exhibits systematic dependence on *both* $N$ and $D$ (Fig. 3 (a) and (b)).

- $\Delta_N L(N, D)$ similarly depends systematically on *both* $N$ and $D$ (Fig. 3 (c) and (d)).

This dual dependence confirms that $L_{ND}(N, D)$ cannot be simplified to depend only on $N$ or only on $D$, nor can it be additively separated. This motivates a fitting approach that can capture such coupled interactions.

## 3.2 Functional Form for Data-Dependent Term $L_D(D) + L_{ND}(N, D)$

This section details the fitting of the data-dependent terms $L_D(D) + L_{ND}(N, D)$, guided by the differential analysis in Sec. 3.1, with the goal of deriving parametric forms for the functions $f_A$ and $f_B$ in Eq. 5. The insights from the differential analysis in Section 3.1 guide the modeling of the data-dependent terms $L_D(D) + L_{ND}(N, D)$. Specifically, the behavior of $\Delta_D L(N, D)$ when plotted against $D$ on log-log axes (Fig. 3 (a)) reveals a consistent and striking linear trend across various model sizes $N$. This indicates a robust power-law relationship for $\Delta_D L(N, D)$ with respect to $D$.

This empirical power-law behavior of the difference term $\Delta_D L(N, D)$ strongly motivates modeling the integrated data-dependent loss component, $L_D(D) + L_{ND}(N, D)$, using a power law in $D$. We therefore propose that this component can be approximated as:

$$L_D(D) + L_{ND}(N, D) = B(N)D^{-A(N)} + \varepsilon_R(N, D) = f_B(N; \theta_B)D^{-f_A(N; \theta_A)} + \varepsilon_R(N, D) \quad (5)$$

The functions $A(N) = f_A(N; \theta_A)$ and $B(N) = f_B(N; \theta_B)$ capture the model-size dependencies of the exponent and coefficient, respectively. Where, $\varepsilon_R(N, D)$ is a stochastic term jointly dependent on $N$ and $D$ that aggregates two sources of variability: the experimental measurement noise inherent to each $(N, D)$ setting and the residual error arising from the fit of the parametric model.

**Stage 1: Initial estimation.** For each unique model size $N$, we first compute the observed quantity of loss difference, it also can be approximated as:

$$R_N(D) = L(N, D) - L(N, \lambda D) \approx B(N)(1 - \lambda^{-A(N)})D^{-A(N)} = \tilde{R}_N(D), \quad (6)$$

where $\lambda = \sqrt{2}$ in experiments. And $\tilde{R}_N(D)$ is predict value of $R_N(D)$ which is projected to log-log space for linear regression:

$$log(\tilde{R}_N(D)) = log(\hat{B}_N) - A_N * log(D), \quad (7)$$

where $B_N = \hat{B}_N/(1 - \lambda^{-A_N})$. Using Normal Equation [15] for minimizing the error of the loss differences of each $N$, respectively, as $\ell_{R,N} = \sum_D (R_N(D) - \tilde{R}_N(D))^2$.

Consequently, for each model size $N$ we obtain a discrete pair of parameters-the exponent $A_N$ and the coefficient $\hat{B}_N$-which constitute the the best linear fit for that particular model size. Collecting these pairs yields the arrays $\{A_N\}$ and $\{B_N\}$.

**Stage 2: Parameterization and Iterative Refinement.** Building on the discrete estimates $\{A_N, B_N\}$ obtained in Stage 1, we now seek continuous functions $f_A(N; \theta_A)$ and $f_B(N; \theta_B)$ that simultaneously (i) admit a compact analytical form and (ii) minimize the global error of the loss differences $\ell_R = \sum_N (\ell_{R,N})$.

Let $\mathcal{G} = \{g_{(1)}, g_{(2)}, \dots\}$ denote a small dictionary of simple, monotone transformations (identity, logarithm, and power functions were found sufficient in practice). For every ordered quadruple $(g_i, g_j, g_k, g_m) \in \mathcal{G}^4$ we perform the *coordinate projection*

$$g_i(A_N) = a_1 g_j(N) + b_1 + \varepsilon_{A,N}, \quad g_k(B_N) = a_2 g_m(N) + b_2 + \varepsilon_{B,N}, \quad (8)$$

where the coefficients $(a_1, b_1)$ and $(a_2, b_2)$ are obtained via the least squares method, yielding best estimates.

For every candidate transform pair we compute the projection-space residual sums of squares $\ell_A = \sum_N \varepsilon_{A,N}^2$ and $\ell_B = \sum_N \varepsilon_{B,N}^2$. The transform quadruple that minimizes $\ell_A + \ell_B$ is selected. As results, $g_i, g_k$ is log function and $g_j, g_m$ is a power-law function, minimized the residuals, implying that both $f_A$ and $f_B$ follow a stretched-exponential form:

$$f_A(N; \theta_A) = \exp(a_1 N^\alpha + b_1), \quad (9)$$

$$f_B(N; \theta_B) = \exp(a_2 N^\beta + b_2). \quad (10)$$

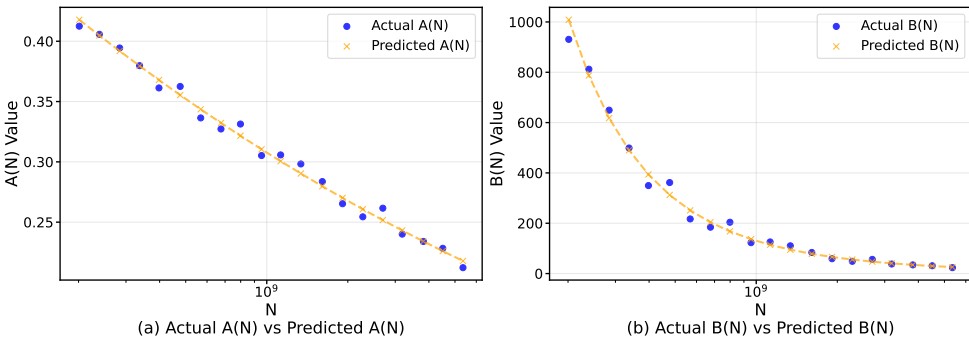

(a) Actual A(N) vs Predicted A(N)          (b) Actual B(N) vs Predicted B(N)

Figure 4: Fits of the terms $A(N)$ (the data scaling exponent) and $B(N)$ (the data scaling coefficient) using stretched exponential functions. Blue dots denote actual values derived from discrete fits, while orange crosses denote predictions from the continuous functions. The close alignment demonstrates the accuracy of the stretched exponential parameterization across a wide range of model sizes $N$.

First, we determine initial parameters for $f_B(N; \theta_B)$ by fitting the function $\log(B_N) = a_2 N^\beta + b_2$ to the discrete data points $\{(N, B_N)\}$ obtained in Stage 1. This provides an initial estimate for the parameter set $\theta_B = (a_2, b_2, \beta)$. An analogous procedure is performed for $f_A(N; \theta_A)$ and its corresponding parameters.

Subsequently, we refine the exponents $\alpha$ and $\beta$ using an *iterative refinement strategy*: (i) holding $\beta$ fixed, we update $\alpha$ to minimize the global residual error $\ell_R$; (ii) holding the new $\alpha$ fixed, we update $\beta$ according to the same criterion. This process is repeated until convergence, which empirically takes only 1-2 iterations. Implementation details, as well as a systematic comparison of alternative functional families and their empirical errors, are provided in Appendix C. Fig 4 illustrates the quality of the resulting fits for $f_A(N)$ and $f_B(N)$ using these stretched exponential forms.

### 3.3 Functional Form for Model-Dependent Residual $E + L_N(N)$

By construction, $E + L_N(N) = L(N, D) - (L_D(D) + L_{ND}(N, D))$, where $L(N, D)$ is directly obtained from experimental data whereas the exact form of $L_D(D) + L_{ND}(N, D)$ is not experimentally accessible. Hence $E + L_N(N)$ cannot be *directly* observed. It can, however, be accessed indirectly through

$$O(N, D) \triangleq L(N, D) - B(N) D^{-A(N)} = E + L_N(N) - \varepsilon_R(N, D), \tag{11}$$

where $\varepsilon_R(N, D)$ is the residual defined in Section 3.2. Grouping by model size $N$ and averaging over $D$ yields

$$G(N) \triangleq \underset{D}{\mathrm{Avg}}[O(N, D)] = E + L_N(N) - \underset{D}{\mathrm{Avg}}[\varepsilon_R(N, D)]. \tag{12}$$

The difference

$$G(N) - O(N, D) = \varepsilon_R(N, D) - \underset{D}{\mathrm{Avg}}[\varepsilon_R(N, D)] \tag{13}$$

serves as a diagnostic of the accuracy of the approximation $B(N)D^{-A(N)}$ to $L_D(D) + L_{ND}(N, D)$. If the fit $B(N)D^{-A(N)}$ accurately approximates $L_D(D) + L_{ND}(N, D)$, the residual variable $\varepsilon_R(N, D)$ should behave like white noise with respect to $N$ and $D$. As shown in Fig. 5 (b), this quantity is distributed as nearly Gaussian white noise across all $N$, with its amplitude shrinking from $2 \times 10^{-3}$ to $4 \times 10^{-4}$ as $N$ increases. Given that $G(N) > 0.21$, these fluctuations are negligible. From Fig. 5 (a), we conclude: (a) the power-law model $B(N)D^{-A(N)}$ provides a sufficiently accurate approximation to $L_D(D) + L_{ND}(N, D)$; (b) the observable $G(N)$ is therefore a reliable estimate of $E + L_N(N)$.

**Fitting $E + L_N(N)$.** We now model $G(N) \approx f_{L_N}(N; \theta_{L_N})$ by repeating the transformation-fit-selection procedure of Section 3.2. For each pair $g_i, g_j \in \mathcal{G}$ we regress $g_i(G(N)) = a_3 g_j(N) + b_3$ via the normal equation and choose the transform pair that minimizes the residual sum of squares The optimal form identified was a logarithmic transformation for $g_i$ and a power-law transformation

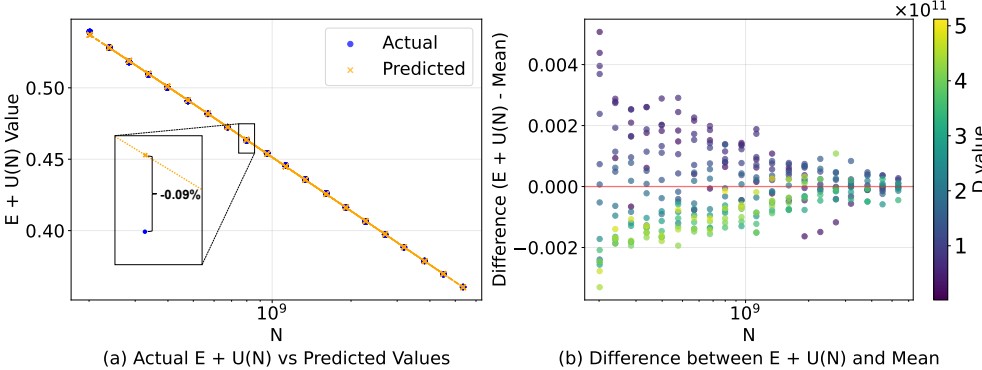

(a) Actual E + U(N) vs Predicted Values          (b) Difference between E + U(N) and Mean

Figure 5: Predicted vs. actual values for the model-dependent residual term $G(N) \approx E + L_N(N)$. (a) Comparison of actual $G(N)$ values (blue dots) and the fitted prediction $\exp(a_3 N^\gamma + b_3)$ (orange crosses) over model size $N$. (b) Residuals $O(N, D) - G(N)$ versus $N$, colored by data size $D$, confirming negligible variance across $D$ and validating $G(N)$ as a reliable estimator for $E + L_N(N)$.

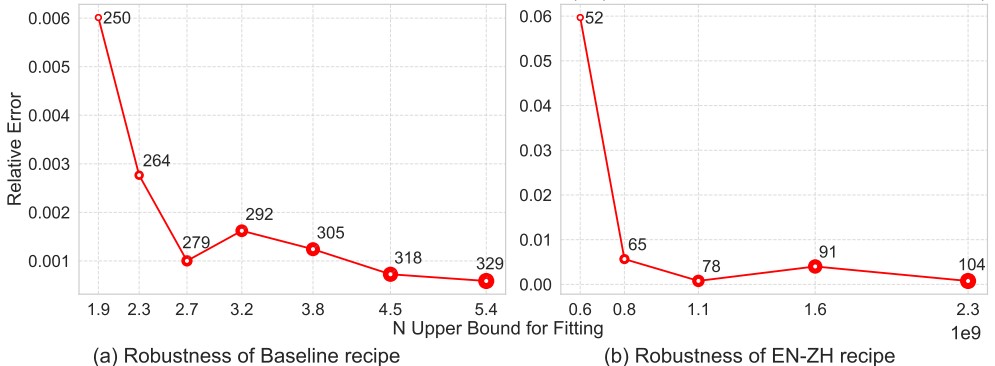

(a) Robustness of Baseline recipe          (b) Robustness of EN-ZH recipe

Figure 6: Robustness and data distribution generalizability of *Farseer*. (a) Relative error on excluded 6.4 B models as a function of the largest model size used in fitting, assessing robustness to fitting data volume. (b) Relative error on excluded 3.2 B models trained with an English-Chinese data recipe, demonstrating structural generalizability to different data mixes. Circle size and adjacent numbers indicate the number of model-size points used for fitting in each case.

$g_j(N) = N^\gamma$ for $g_j$. The optimal exponent $\gamma$ is identified via a grid search minimizing the fitting error, leading to a stretched-exponential form

$$E + L_N(N) \; \simeq \; G(N) \; \approx \; \exp\!\big(a_3\, N^\gamma + b_3\big). \tag{14}$$

Fig. 5 (a) shows that this fit attains a relative error of only $0.09\%$. Given the negligible estimation error of $G(N)$ in approximating $E + L_N(N)$, we have thus completed a robust modeling and fitting of the model-dependent residual. Combining these stages yields the fully specified scaling law in Eq. 3, with all functional forms and parameters fixed. Further details, ablation studies, and noise analyses are provided in Appendix C.

## 4  *Farseer*'s Properties: Robustness, Generalization, and Extrapolation

This section evaluates key properties of *Farseer*: its **robustness** to fitting data amount, its **generalization** across different training data recipes, and its **extrapolation** ability beyond the fitting range.

### 4.1  Robustness to Fitting Data

We measure how prediction accuracy changes with data size to assess *Farseer*'s robustness, parameter stability, and prediction consistency as more model-size points become available. Specifically, we fit *Farseer* using subsets of models with progressively increasing upper bounds on model size $N$. For each fitting process based on a subset, we consistently measure the relative error on the 6.4

B models, which is deliberately excluded from all fitting subsets. As illustrated in Fig 6 (a), the predicted relative error on the excluded 6.4 B models decreases significantly as more data points are included in the fit. When fitting only models up to 1.9 B parameters, the relative error is $6.01 \times 10^{-3}$; expanding the fit to include models up to 5.4 B reduces it to only $5.87 \times 10^{-4}$. Crucially, the relative error decreases nearly monotonically as additional model-size points are incorporated. This trend highlights *Farseer*'s accuracy and robustness to fitting-data volume, yielding reliable predictions with limited data and predictable gains as more data are added.

## 4.2 Data Distribution Generalization

To investigate the generalizability of *Farseer*, we evaluate it on a different training data setup. All prior experiments used the Baseline data recipe, specified in Appendix B. Fig 6 (b) extends this analysis to evaluate *Farseer* using data from models trained with the English-Chinese (EN-ZH) recipe (also detailed in Appendix B), which represents a dramatically different bilingual data mix. Similar to the robustness analysis, as the fitting data increases, the relative error on deliberately excluded 3.2 B models trained with the EN-ZH recipe steadily converges and stays consistently low, reaching $7.60 \times 10^{-4}$ at the final point. This shows that *Farseer* captures key scaling trends and delivers reliable predictions even under a dramatically altered bilingual data mix. This result underscores *Farseer*'s structural generalizability across diverse training-data compositions and suggests its applicability to models trained on varied datasets. As an illustration, a detailed surface comparison of different English-Chinese data mixture ratios is provided in Appendix E.

## 4.3 Extrapolation Capabilities

To quantify the extrapolative capacity of *Farseer*, we fit its parameters on a $\sqrt{2}$-spaced sampling grid of small model sizes and dataset sizes, then predict BPC values for substantially larger and off-grid combinations. Fig. 7 presents six such extrapolation targets (red stars) that situate well outside the calibration region (blue circles). Most notably, the 25.1 B model extends the evaluation domain beyond the largest calibrated scale by more than an order of magnitude. Despite this, *Farseer*'s prediction for this model exhibits a relative error of merely 0.47 %. The remaining validation points selected to assess extrapolation at both increased dataset sizes and off-grid $(N, D)$ configurations demonstrate similarly low relative errors, ranging from 0.26 % to 0.72 %. Across these extrapolation targets, **_Farseer_'s average relative error is just 0.50 %, whereas the Chinchilla scaling law exhibits an average relative error of 2.68 %, a 433 % increase.**

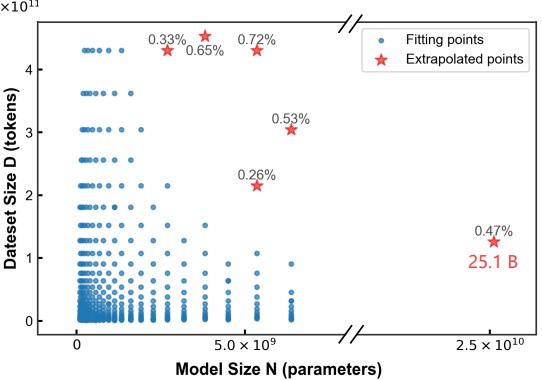

Figure 7: Extrapolation of *Farseer*. Blue circles represent the grid of $(N, D)$ employed to fit. Red stars denote validation points beyond that distribution, including a 25.1 B model, larger dataset sizes, and off-grid combinations. Annotated percentages give the relative errors of each extrapolated point.

These consistently low errors demonstrate that *Farseer* accurately captures the smooth functional dependence of BPC on both model and dataset scale. In particular, the highly accurate extrapolation at the 25.1 B model provides compelling evidence that *Farseer* generalizes robustly to previously unfitted scales, thereby serving as a reliable instrument to forecast the performance under yet larger computational budgets.

## 4.4 Formula and Monotonicity

As formulated in Eq. (3) and derived via our fitting method, *Farseer* possesses intrinsic mathematical properties that align with the theoretical expectations for loss functions in machine learning. This section details the fitted formula and its fundamental property of monotonicity.

**Formula.** The specific fitted law for *Farseer* is given by:

$$L(N, D) = e^{-0.021 \cdot N^{0.169} - 0.091} + e^{88.01 \cdot N^{-0.1} - 6.287} \cdot D^{-e^{-0.124 \cdot N^{0.123} + 0.424}} \tag{15}$$

**Monotonicity.** A fundamental expectation is that model performance should improve (i.e., loss should decrease) with more resources. *Farseer* inherently satisfies this: $L(N, D)$ is monotonically decreasing with respect to both increasing $N$ and $D$. This behavior is a direct consequence of its functional form and the parameter constraints imposed by our fitting procedure. A detailed analysis of the partial derivatives confirming this monotonicity is provided in Appendix I.

## 5 Comparative Analysis of *Farseer* and Chinchilla

We compare our proposed scaling law *Farseer* against the widely recognized Chinchilla [20] in Eq. (2). This comparison highlights the advantages of our approach in terms of predictive accuracy, extrapolation, and guidance on optimal resource allocation.

### 5.1 Formula Comparison: Predictive Power

We compared the predictive power of Chinchilla and Farseer using our comprehensive dataset (Appendix B). Standard non-linear regression was used for fitting both, and Chinchilla was also evaluated with our multi-round iterative method (Appendix H). As shown in Fig. 1 (a) and Fig. 2, *Farseer*'s predicted BPC aligns remarkably closely with empirical values across various $N$ and $D$. Conversely, Chinchilla fit systematically deviates, underestimating at low $D$ and overestimating at high $D$. This superior fit underscores *Farseer*'s higher expressive capacity, enabled by its $A(N)$ and $B(N)$ that capture decay rates specific to each model size, unlike Chinchilla's single average trend.

### 5.2 Property Comparison: Robustness and Extrapolation

To assess the robustness and extrapolation, we examine the generalization beyond the training domain in two scenarios (see Fig. 8), plotting relative error as the upper bound on model size $N$ used for fitting increases. As shown in Fig. 8 (a), with model size fixed at 6.4 B parameters, we vary $N$ from $1.9 \times 10^9$ to $5.4 \times 10^9$ and observe that *Farseer* formula yields both lower average relative error and smaller error variance than the Chinchilla fit, indicating enhanced robustness to changes in the fitting range. Fig. 8 (b) considers a fixed model size of 25.1 B, which is far beyond the fitting range. As $N$ increases, *Farseer*'s relative error shows a clear downward trend and reaches a low level, while Chinchilla's error remains persistently high. These results collectively demonstrate that *Farseer* not only offers greater robustness within the fitting range but also generalizes more reliably when extrapolating to larger model scales.

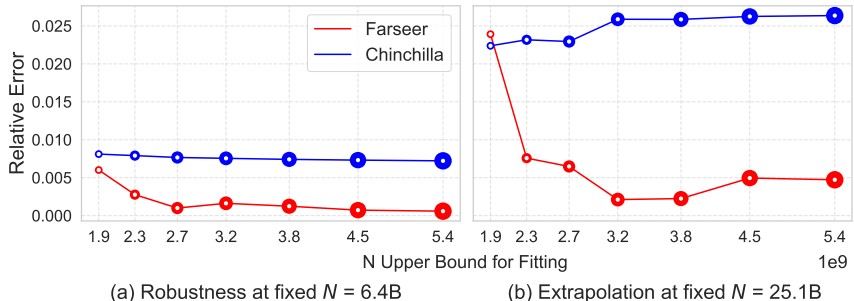

Figure 8: *Farseer* vs Chinchilla at robustness and extrapolation. The x-axis is "Largest model size used in fitting". (a) For 6.4 B models, *Farseer* shows lower and steadier errors as $N$ upper bound increases. (b) For 25.1 B model, far beyond the fitting range, *Farseer* achieves a clear error reduction as $N$ increases, while Chinchilla stays large.

### 5.3 Application Comparison: Optimal Computing Resource Allocation

Efficient training requires balancing model size $N$ and dataset size $D$, typically reflected in the optimal $D/N$ ratio. we compare the practical guidance offered by *Farseer* and Chinchilla on optimal compute allocation during training, where the total budget is taken as $C = 6ND$ [22, 20]. Fig. 1 (b) presents the optimal $D/N$ ratio predicted by both *Farseer* and Chinchilla as a instrument of the total compute budget $C$. Chinchilla, suggesting an optimal $D/N$ ratio around 20 (typically 10-30), is primarily derived from and applicable to training at moderate compute budgets corresponding to smaller model

sizes. This constant ratio provides inaccurate guidance for larger scale. In contrast, *Farseer* predicts a steadily increasing optimal $D/N$ ratio as the compute budget (and consequently, the optimal model size) grows. This predicted trend aligns remarkably well with the actual training configurations adopted for recent state-of-the-art large language models (e.g., Qwen [46, 45], Llama [39, 40, 16]), indicating that *Farseer* offers more accurate and relevant guidance for optimizing resource allocation in current and future large-scale model training.

## 6 Related Work

Training LLMs necessitates optimizing compute allocation via scaling laws. The discovery and application of these laws have provided the confidence to invest massive compute resources, knowing they can be predictably converted into model capability [22]. OpenAI's foundational work established a power-law relationship between cross-entropy loss and model/data scale (Eq. 1), advocating for training large models on moderate data with early stopping. However, this assumes oversimplified dynamics and fixed hyperparameters, limiting extrapolation reliability.

DeepMind's Chinchilla [20] refined this by proposing proportional scaling of model and data size based on an updated methodology (Eq. 2), empirically demonstrating improved efficiency. Under the same compute budget as Gopher [32], Chinchilla adopts a smaller model paired with more training data, yielding substantial gains on downstream tasks. Nevertheless, recent work [4] has raised concerns about the reproducibility of Chinchilla's findings and demonstrated that optimal token-to-parameter ratios are sensitive to data quality, suggesting current scaling laws are valuable heuristics but require calibration and lack universal applicability.

The landscape of scaling law research is rapidly evolving beyond classical formulations, exploring multifaceted avenues for enhanced efficiency and generalization. Recent empirical work includes deriving observational scaling laws from existing models to unify performance patterns and explain emergent phenomena [33, 3, 18, 42], while other studies apply scaling principles to improve data curation, *e.g.*, by strategically filtering datasets [27, 26, 35, 30, 48, 13]. Theoretical inquiries are concurrently advancing our comprehension of core mechanisms: some demonstrate how sophisticated feature learning can potentially double scaling rates for intricate functions [5], while others establish direct connections between generalization error exponents and data manifold dimensionality [18]. Furthermore, the purview of scaling laws now extends significantly beyond pre-training, with investigations into their implications for inference dynamics [43, 44, 24, 43, 34], parameter and communication efficiency [1, 8], and downstream task [21, 9, 44, 19, 36, 10, 12, 37, 38, 28, 23].

## 7 Limitations

This study has several limitations. First, our empirical validation is primarily based on Llama-style decoder-only Transformers. While the core methodology may be generalizable, its applicability to other architectures, such as Mixture-of-Experts (MoE), requires further investigation. Second, our largest validated model has 25.1B parameters. Extrapolating to trillion-parameter models remains an open question, constrained by computational resources and the engineering complexities of large-scale parallel training. Third, in line with other work in this area, our scaling law is empirically derived and lacks a first-principles theoretical justification. We have open-sourced our data to encourage community efforts on this front. Finally, this work focuses on pre-training loss (BPC); extending scaling laws to predict downstream task performance is a complex but important direction for future work. A more detailed discussion of limitations is available in Appendix J.

## 8 Conclusion

We propose *Farseer*, a refined scaling law and methodology for Large Language Models. By accurately modeling the loss surface $L(N, D)$ through a novel fitting approach, *Farseer* provides significantly better empirical fit and superior extrapolation capabilities compared to prior scaling laws like Chinchilla's. This work enables reliable prediction of large-scale model performance from tractable small-scale experiments, bridging the critical scaling gap and facilitating more efficient evaluation of training strategies and compute allocation. Validated on a large corpus of trained models, *Farseer* offers valuable insights for LLM development.

## Acknowledgments

This research is supported in part by the National Key Research and Development Program of China under Grant 2023ZD0121300, and in part by the Natinonal Natural Scinece Foundation of China(Grant No. 62495092).

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

# Appendix

## Content

# A  General Loss Function Metric Formulas

To systematically evaluate and predict the scalability of various LLM training approaches, we introduce a general loss formulation expressed as a loss surface $L(N, D)$. This surface captures the model's performance metrics (BPC), as a function of model size ($N$) and training data size ($D$). The loss surface $L(N, D)$ is implicitly conditioned on a fixed underlying training strategy, serving as a foundational tool for scalability analysis.

To formalize our analysis, we define a specific Large Language Model (LLM) instance as $\mathbb{M} \triangleq (\mathbb{A}, N)$. Here, $\mathbb{A}$ denotes a *LLM family*, characterizing models that share common architectural features or design principles, such as Transformer decoder-only architectures with fixed aspect ratios. Given a parameter count $N$, a deterministic LLM instance $\mathbb{M}$ is realized from the family $\mathbb{A}$.

The performance of an LLM instance, determined through a specific training regimen and measured by metrics, depends on intrinsic model properties (e.g., $N$) and various training-related factors. We formalize this aggregate performance metric $L$ as a function:

$$L(\mathbb{A}, \mathbb{D}, N, D, \text{LR}, \text{BS}, \delta) \tag{16}$$

where:

- $\mathbb{A}$: The LLM family.
- $\mathbb{D}$: The dataset or data distribution used for training.
- $N$: The parameter count of the model instance.
- $D$: A measure of the training data scale (e.g., number of tokens).
- $\text{LR}, \text{BS}$: The learning rate and batch size used during training.
- $\delta$: A term encapsulating other influential hyperparameters or training specifics, such as optimizer type, regularization strategies, or training duration.

Furthermore, in studying the Scaling Law, we assume that other hyperparameters (e.g., learning rate, batch size) are set to *appropriate values* that prevent the model training from deviating excessively or performing too poorly, or they follow scale-aware heuristics [47, 25]. Under this assumption, performance $L$ becomes primarily a function of $N$ and $D$, and our central objective is to precisely characterize the loss surface $L(N, D)$ for a given model family $\mathbb{A}$ and data recipe $\mathbb{D}$.

As established in 2.1, our primary objective is to accurately characterize and predict the loss surface $L(N, D)$, which describes how performance scales with model size ($N$) and data size ($D$) for a specific model family $\mathbb{A}$ and data recipe $\mathbb{D}$. However, the overall performance metric $L$, as indicated by our general formulation (Eq. (16)), is also sensitive to factors such as learning rate ($lr$), batch size ($bs$), and specific architectural choices within the family $\mathbb{A}$.

To ensure a stable and deterministic relationship between the primary variables $(N, D)$ and the performance $L$, thereby enhancing the robustness and accuracy of subsequent scaling law fitting, it is crucial to control for or standardize these additional sources of variation. This typically involves preliminary experiments and adherence to fixed protocols.

In this study, we utilize a fixed data recipe $\mathbb{D}$ (details on its construction and characteristics are deferred to table 1). Consequently, the main efforts in controlling variability focus on standardizing the following two categories of factors:

1. **Optimization Hyperparameters:** Primarily the learning rate ($lr$) and batch size ($bs$), which are set to suitable values (potentially following scale-aware schedules or determined through preliminary sweeps) to ensure model training does not deviate significantly or exhibit markedly poor performance.

2. **Architectural Hyperparameters:** The specific configuration defining the LLM family $\mathbb{A}$ (e.g., layer counts, hidden dimensions relative to $N$, activation functions) must be consistently defined or scaled according to precise rules.

By carefully controlling these factors, we ensure that for a given experimental setup defined by $\mathbb{A}$ and $\mathbb{D}$ and the chosen hyperparameter protocols, each $(N, D)$ pair maps to a well-defined expected performance value $L$. This satisfies a fundamental prerequisite for reliably fitting the loss surface $L(N, D)$ and accurately assessing scaling behavior.

## A.1 Standardizing Hyperparameter Settings

We first address the influence of training hyperparameters, primarily learning rate ($lr$) and batch size ($bs$), on performance $L$. Prior work [25] and our preliminary investigations confirm their significant impact. Crucially, for a given model scale ($N$) and data size ($D$), while one might pursue strictly optimal hyperparameters ($lr_{\text{opt}}(N, D), bs_{\text{opt}}(N, D)$), our actual requirement is for *appropriate* hyperparameters. These are settings that ensure the model training does not deviate excessively or perform markedly poorly, rather than needing to be strictly optimal. Furthermore, the performance landscape $L$ is often relatively flat near regions of good hyperparameter choices.

This flatness implies that using such appropriate hyperparameters yields stable and sufficiently good results, effectively mitigating performance fluctuations due to minor hyperparameter variations, without the need to pinpoint strict optima. Since the goal of scaling law research is to understand model potential under competent, rather than necessarily strictly optimized, training conditions, we operate by selecting hyperparameters—primarily determined by $N$ and $D$ for a fixed model family $\mathbb{A}$ and data recipe $\mathbb{D}$—that ensure they are adequate for effective training and do not cause the model to perform substantially worse than it otherwise could. This allows us to simplify the loss function, treating $L$ primarily as a function of $N$ and $D$ by implicitly using these chosen, appropriate hyperparameters (denoted $lr_{\text{appr}}, bs_{\text{appr}}$):

$$L_{\mathbb{A},\mathbb{D}}(N, D, lr, bs) \approx L_{\mathbb{A},\mathbb{D}}(N, D, lr_{\text{appr}}, bs_{\text{appr}}) \equiv L_{\mathbb{A},\mathbb{D}}(N, D)$$

In practice, we leverage methods like the Step Law [25] to help select suitable $lr$ and $bs$ values for our experiments, applying these standardized settings consistently across different $(N, D)$ points to ensure model performance is not significantly hindered by hyperparameter selection.

## A.2 Defining an Architecturally Standardized LLM Family

The concept of an LLM family provides a robust framework for systematic analysis and comparison. In a broad sense, a family can encompass models sharing a common set of training methods, data processing pipelines, or fundamental architectural paradigms. The core idea is that each distinct family is hypothesized to follow its own unique scaling law. By first defining a family and then empirically determining its scaling behavior, we can create a principled basis for comparing the efficiency and potential of different approaches—for instance, to see which family scales more effectively with increased compute or data.

We define an LLM family such that, given a parameter budget $N$, one can uniquely determine every architectural detail of the model: including, for example, the aspect ratio, the FFN ratio, the type of attention mechanism (multi-head attention or group-query attention), whether the network is pre-norm or post-norm, the activation function used in the FFN, the head dimension/number of heads, and whether a mixture-of-experts (MoE) architecture is employed.

With this definition, models of different sizes within the same LLM family share the same scaling-law characteristics. Consequently, once an LLM family $\mathbb{A}$ and a data recipe $\mathbb{D}$ are fixed, they uniquely determine a loss surface $L_{\mathbb{A},\mathbb{D}}(N, D)$. Under the same data distribution, two different LLM families yield two distinct surfaces—$L_{\mathbb{A}_1,\mathbb{D}}(N, D)$ and $L_{\mathbb{A}_2,\mathbb{D}}(N, D)$—which allows us to compare how architectural choices affect scaling behaviour. Similarly, for a fixed LLM family trained on different data distributions, the respective surfaces $L_{\mathbb{A},\mathbb{D}_1}(N, D)$ and $L_{\mathbb{A},\mathbb{D}_2}(N, D)$ reveal how data characteristics influence the scaling law.

**Example.** Even within a fixed architecture—for instance, a LLaMA-style dense LLM (pre-norm + MHA)—variations in model shape can still impact performance. We take two critical structural ratios as examples that govern model shape and are known to influence performance and computational efficiency:

- **Aspect Ratio:** The ratio of model width to depth ($d_{\text{model}}$ / Number of Layers).
- **FFN Ratio:** The ratio of the feed-forward network's intermediate size to the hidden dimension ($d_{\text{ffn}}/d_{\text{model}}$).

We conducted controlled experiments to study the independent effects of these ratios on performance $L$, holding either total parameters or estimated computational cost approximately constant. As shown in Fig. 9, our results consistently indicate that performance exhibits a relatively flat optimum

with respect to both aspect ratio and FFN ratio across different experimental constraints. Motivated by the patterns observed in Fig. 9, all experiments in this paper therefore fix both the aspect ratio and the FFN ratio. The concrete values we adopt lie in the sub-optimal plateau region identified in the figure. This example illustrates that when using FARSEER to define a custom LLM family $\mathbb{A}$, one must not only keep the architectural blueprint unchanged, but also hold model-shape-related hyperparameters fixed. These configuration variables need not be globally optimal, but they must remain constant as $N$ varies.

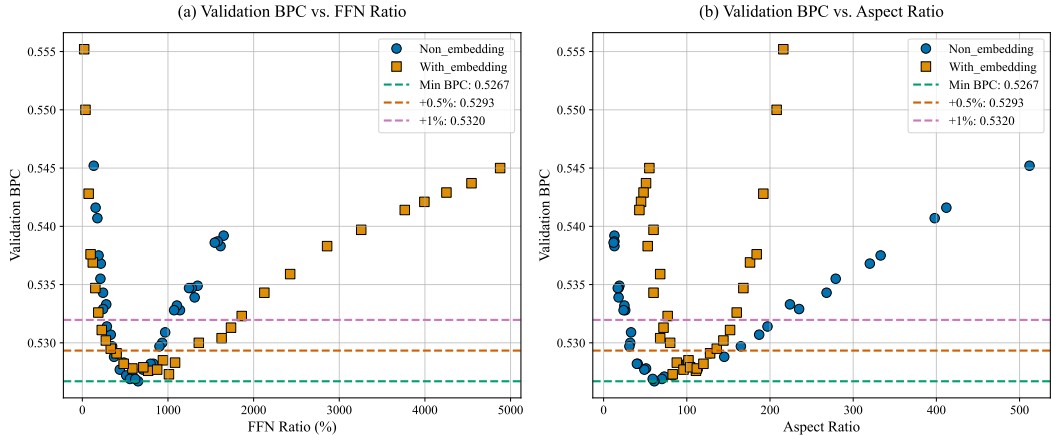

Figure 9: Validation BPC (see appendix D) under a fixed parameter $N$ versus (a) FFN ratio (intermediate size / hidden dimension) and (b) aspect ratio (hidden dimension / layer count). Blue circles: no embeddings; orange squares: with embeddings. Dashed lines mark min BPC (green), +0.5% (orange) and +1% (purple). Both curves exhibit a broad, flat optimum.

## B    Distribution of the Training Dataset

This appendix details the composition of the data recipes used for training our models. Each recipe represents a specific mix of datasets, weighted to target distinct capabilities. All prior experiments utilized the Baseline data recipe. Furthermore, to evaluate *Farseer* under a dramatically different bilingual data mix, experiments were also conducted using data from models trained with the English-Chinese (EN-ZH) recipe. The table below specifies the percentage weights of the constituent datasets for these two primary recipes.

Table 1: **Distribution of dataset weights (in percent) across various training strategies**. Each strategy targets a distinct capability: baseline performance, enhanced coding and mathematical reasoning, or English-Chinese bilingual proficiency.

| Dataset | Baseline | EN-ZH |
|---|---|---|
| web-data-en | 79.53 | 44.99 |
| web-data-cn | – | 34.52 |
| code-the-stack | 4.62 | 4.63 |
| web-data-math | – | – |
| book-non-novel-en | 4.35 | 4.35 |
| paper | 3.38 | 3.38 |
| wikipedia-mtlg | 3.24 | 3.25 |
| stackexchange | 2.21 | 2.22 |
| wikipedia-en | 1.69 | 1.69 |
| book-novel-en | 0.83 | 0.83 |
| wikipedia-cn | 0.13 | 0.13 |

# C  Ablation Study of Fitting Methods

## C.1  Algorithm Workflow: Differential Piecewise Fitting

The Differential Piecewise Fitting procedure, detailed in Algorithm 2, systematically models data $L(N, D)$. Many steps in this procedure involve fitting functional forms, for example, to model how parameters $A_N$ and $B_N$ depend on $N$. Algorithm 1 describes a general method, Optimal Transformation Selection, for choosing appropriate data transformations (e.g., logarithmic, power-law) from a dictionary $\mathcal{G}$ to linearize relationships and subsequently fit parameters. The overall Differential Piecewise Fitting procedure, as implemented in Algorithm 2 using specific stretched-exponential forms, comprises three main stages:

1. **Stage 1: Initial Estimation of $A_N$ and $B_N$:** For each model size $N$, the parameters $A_N$ and $B_N$ of the data-dependent term $B(N)D^{-A(N)}$ are estimated by analyzing the finite difference $\Delta_D L(N, D)$, following the specific regression steps detailed in Algorithm 2.

2. **Stage 2: Parameterization and Iterative Refinement of $f_A(N; \theta_A)$ and $f_B(N; \theta_B)$:** Continuous functions $f_A(N; \theta_A)$ and $f_B(N; \theta_B)$ are derived by fitting the discrete $\{A_N\}$ and $\{B_N\}$ estimates. While Algorithm 2 specifies particular functional forms (stretched-exponentials), the underlying task of selecting optimal transformations and fitting parameters is generally addressed by the methodology in Algorithm 1. This stage also includes iterative refinement of power-law exponents (e.g., $\alpha, \beta$) as specified in Algorithm 2.

3. **Stage 3: Fitting the Model-Dependent Residual $E + L_N(N)$:** The residual term $E + L_N(N)$ is estimated (typically by averaging $L(N, D) - B(N; \theta_B^*)D^{-A(N; \theta_A^*)}$ over $D$ to get $G(N)$) and modeled as $f_{L_N}(N; \theta_{L_N})$. Similar to Stage 2, Algorithm 2 employs a specific stretched-exponential functional form for $f_{L_N}(N)$, and the transformation selection and fitting process to determine its parameters can be understood in the general context of the methods presented in Algorithm 1.

---

**Algorithm 1:** Optimal Transformation Selection for $Y = f(X)$

---

**Input:** Discrete data points $(X_k, Y_k)$ for $k = 1, \ldots, M$.
**Input:** Dictionary of candidate transformation functions $\mathcal{G} = \{g_{(1)}, g_{(2)}, \ldots\}$.
// Each $g \in \mathcal{G}$ is a function, e.g., identity, logarithm.
**Output:** Optimal transformations $g_Y^*, g_X^* \in \mathcal{G}$.
**Output:** Coefficients $(a^*, b^*)$ for the linear model $g_Y^*(Y_k) \approx a^* g_X^*(X_k) + b^*$.
**Output:** Minimum residual sum of squares $\ell_{\min}$.

---

1   Initialize $\ell_{\min} \leftarrow \infty$
2   Initialize $g_Y^* \leftarrow$ null, $g_X^* \leftarrow$ null, $a^* \leftarrow$ null, $b^* \leftarrow$ null, $p^* \leftarrow$ null
3   **forall** $g_Y \in \mathcal{G}$ **do**
4      **forall** $g_X \in \mathcal{G}$ **do**
        // Define $Y_k' = g_Y(Y_k)$. Define $X_k' = g_X(X_k)$.
5         Perform linear regression: $Y_k' = a_{cand}X_k' + b_{cand}$
6         Calculate current residual sum of squares $\ell_{\text{current}} \leftarrow \sum_k (Y_k' - (a_{cand}X_k' + b_{cand}))^2$
7         **if** $\ell_{current} < \ell_{min}$ **then**
8            $\ell_{\min} \leftarrow \ell_{\text{current}}$
9            $(g_Y^*, g_X^*) \leftarrow (g_Y, g_X)$
10           $(a^*, b^*) \leftarrow (a_{cand}, b_{cand})$

11   **return** $g_Y^*, g_X^*, (a^*, b^*), \ell_{\min}$

---

**Algorithm 2:** Differential Piecewise Fitting (with Stretched-Exponential Forms)

**Input:** Loss Data points $L(N, D)$, scale factor $\lambda$.
**Output:** Parameters $\theta_A^* = (a_1^*, b_1^*, \alpha^*)$, $\theta_B^* = (a_2^*, b_2^*, \beta^*)$, $\theta_{L_N}^* = (a_3^*, b_3^*, \gamma^*)$.

**Output:** Final fit $L(N, D) \approx \exp(a_2^* N^{\beta^*} + b_2^*) D^{-\exp(a_1^* N^{\alpha^*} + b_1^*)} + \exp(a_3^* N^{\gamma^*} + b_3^*)$.

```
// Stage 1: Initial estimation of discrete A_N, B_N
```
1 **foreach** *model size $N$* **do**
2     Compute $R_N(D) \leftarrow L(N, D) - L(N, \lambda D)$
```
   // From text: R_N(D) ≈ B̂_N D^{-A_N}
```
3     Estimate $A_N, \hat{B}_N$ via linear fit on
4        $\log(R_N(D)) = \log(\hat{B}_N) - A_N \log(D)$
```
   // Linear fit parameters can be found using the Normal Equation.
```
5     $B_N \leftarrow \hat{B}_N / (1 - \lambda^{-A_N})$
6 Collect discrete sets $\{A_N\}$ and $\{B_N\}$

```
// Stage 2: Parameterization and Iterative Refinement of f_A(N;θ_A) and f_B(N;θ_B)
// Assumed forms: f_A(N;θ_A) = exp(a_1 N^α + b_1),  f_B(N;θ_B) = exp(a_2 N^β + b_2).
```
7 Fit $\log(B_N) = a_2 N^\beta + b_2$ to $\{(N, B_N)\}$ to find initial $a_2, b_2, \beta$
```
   // Similarly, for each β, (a_2, b_2) are found via linear regression (e.g., Normal
        Equation) minimizing ∑_N (log(B_N) - (a_2 N^β + b_2))².
```

```
// Iterative refinement of exponents α, β
```
8 Let $\tilde{R}_N(D; \theta_A, \theta_B) = f_B(N; \theta_B)(1 - \lambda^{-f_A(N;\theta_A)}) D^{-f_A(N;\theta_A)}$
9 Let global residual error $\ell_R = \sum_N \sum_D (R_N(D) - \tilde{R}_N(D; \theta_A, \theta_B))^2$
10 **repeat**
11     Fix $\beta, a_2, b_2$. Update $\alpha$ (and re-estimate $a_1, b_1$) to minimize $\ell_R$
```
   // This involves finding α (e.g., via grid search). For each candidate α, (a_1, b_1)
        are re-calculated by fitting log(A_N) = a_1 N^α + b_1. The set (α, a_1, b_1) that
        minimizes the global residual ℓ_R is chosen.
```
12     Fix updated $\alpha, a_1, b_1$. Update $\beta$ (and re-estimate $a_2, b_2$) to minimize $\ell_R$
```
   // Similarly, this involves finding β. For each candidate β, (a_2, b_2) are
        re-calculated by fitting log(B_N) = a_2 N^β + b_2. The set (β, a_2, b_2) that minimizes
        the global residual ℓ_R is chosen.
```
13 **until** *convergence (e.g., 1-2 iterations or small change in $\ell_R$)*
14 Obtain refined parameters $\theta_A^* = (a_1^*, b_1^*, \alpha^*)$ and $\theta_B^* = (a_2^*, b_2^*, \beta^*)$ from the best fit

```
// Stage 3: Fit model-dependent residual E + L_N(N)
// Assumed form: f_{L_N}(N;θ_{L_N}) = exp(a_3 N^γ + b_3).
```
15 Compute $O(N, D) \leftarrow L(N, D) - f_B(N; \theta_B^*) D^{-f_A(N;\theta_A^*)}$
16 $G(N) \leftarrow \text{Avg}_D[O(N, D)]$
17 Fit $\log(G(N)) = a_3 N^\gamma + b_3$ to $\{(N, G(N))\}$ to find $a_3, b_3, \gamma$
```
   // This involves finding γ (e.g., grid search); for each γ, (a_3, b_3) are found via
        linear regression (e.g., Normal Equation) minimizing ∑_N (log(G(N)) - (a_3 N^γ + b_3))².
```
18 Obtain final parameters $\theta_{L_N}^* = (a_3^*, b_3^*, \gamma^*)$

```
// Final fitted Scaling Law
```
19 The final scaling law is:
20

$$L(N, D) \approx \exp(a_2^* N^{\beta^*} + b_2^*) D^{-\exp(a_1^* N^{\alpha^*} + b_1^*)} + \exp(a_3^* N^{\gamma^*} + b_3^*)$$

**return** $\theta_A^*, \theta_B^*, \theta_{L_N}^*$

### C.2 Core Insight: Power-Law Scaling with Data Size ($D$)

The entire methodology hinges on a key empirical observation: the finite difference of the loss with respect to data size, $\Delta_D L(N, D) = L(N, \lambda D) - L(N, D)$, exhibits a robust power-law relationship with $D$ when plotted on log-log axes. This is illustrated in Figures 12, 13, and14, which shows a

striking linear trend for $\Delta_D L(N, D)$ vs. $D$ across various model sizes $N$. The left panel shows the power-law fit on a log-log scale, while the right panel shows the relative residuals of the fit, which are small and randomly distributed, indicating a good fit. This observation is consistent across all studied ranges of model size $N$, achieving a high average $R^2$ value of 0.9807.

This observation, $\Delta_D L(N, D) \propto D^{-A(N)}$, constitutes the **central empirical insight** derived directly from the data analysis concerning the functional form of the data-dependent loss components $L_D(D) + L_{ND}(N, D)$. This insight allows us to approximate $L_D(D) + L_{ND}(N, D) \approx B(N) D^{-A(N)}$, where $A(N)$ and $B(N)$ capture model-size dependencies. The cancellation of $E$ and $L_N(N)$ terms in $\Delta_D L(N, D)$ simplifies the analysis, focusing it on data-dependent effects. This power-law relationship is pivotal, as all subsequent fitting stages for $A(N)$, $B(N)$, and consequently $L_N(N)$, build upon this initial characterization.

To establish the most reliable functional forms for the components of the loss $L(N, D)$, we analyzed various differential perspectives. The general form of the loss is $L(N, D) = L_D(D) + L_{ND}(N, D) + E + L_N(N)$. We consider two primary types of finite differences:

- $\Delta_D L(N, D) = L(N, D) - L(N, \lambda D) \approx [L_D(D) - L_D(\lambda D)] + [L_{ND}(N, D) - L_{ND}(N, \lambda D)]$
- $\Delta_N L(N, D) = L(N, D) - L(\lambda N, D) \approx [L_N(N) - L_N(\lambda N)] + [L_{ND}(N, D) - L_{ND}(N, \lambda D)]$

Each of these differences can be analyzed as a function of $N$ (with $D$ fixed) or $D$ (with $N$ fixed), potentially revealing power-law relationships. This gives four primary perspectives to model the interaction term $L_{ND}(N, D)$ and its associated main effects:

1. $\Delta_D L(N, D)$ **vs. $D$ (for fixed $N$):** Leads to fitting $\hat{B}_D(N) D^{-A_D(N)}$. The terms $A_D(N)$ and $\hat{B}_D(N)$ are coefficients and exponents that depend on $N$.

2. $\Delta_D L(N, D)$ **vs. $N$ (for fixed $D$):** Leads to fitting $\hat{E}_N(D) N^{-C_N(D)}$. Here, $\hat{E}_N(D)$ and $C_N(D)$ are coefficients and exponents that depend on $D$.

3. $\Delta_N L(N, D)$ **vs. $D$ (for fixed $N$):** Leads to fitting $\hat{G}_D(N) D^{-F_D(N)}$. Here, $\hat{G}_D(N)$ and $F_D(N)$ are coefficients and exponents that depend on $N$.

4. $\Delta_N L(N, D)$ **vs. $N$ (for fixed $D$):** Leads to fitting $\hat{I}_N(D) N^{-H_N(D)}$. Here, $\hat{I}_N(D)$ and $H_N(D)$ are coefficients and exponents that depend on $D$.

Figures 10 and 11 illustrate the behavior of the coefficient and exponent functions derived from these four perspectives.

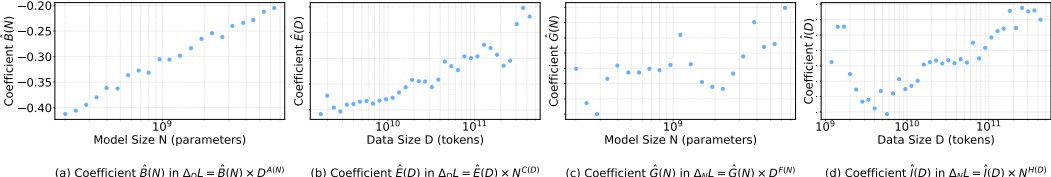

Figure 10: Log-Log projections of the coefficient functions derived from four differential loss perspectives: (a) $\hat{B}(N)$ in $\Delta_D L = \hat{B}(N) D^{-A(N)}$ vs. $N$; (b) $\hat{E}(D)$ in $\Delta_D L = \hat{E}(D) N^{-C(D)}$ vs. $D$; (c) $\hat{G}(N)$ in $\Delta_N L = \hat{G}(N) D^{-F(N)}$ vs. $N$; (d) $\hat{I}(D)$ in $\Delta_N L = \hat{I}(D) N^{-H(D)}$ vs. $D$. Perspective (a), $B(N)$ vs. $N$, exhibits the most consistent and regular scaling, motivating its use in the primary analysis.

As indicated in Figure 10(a), the coefficient $B(N)$ (derived from $\Delta_D L(N, D)$ vs. $D$) displays an exceptionally tight and consistent scaling with $N$. Similarly, Figure 11(a) shows that the corresponding exponent $A(N)$ varies smoothly and predictably over a wide range of $N$. This regularity and stability are significantly more pronounced than those observed for analogous coefficient and exponent terms derived from the other three perspectives (Figures 10(b-d) and 11(b-d)).

The superior numerical behavior and clearer trends of $A(N)$ and $B(N)$ obtained from the $\Delta_D L(N, D)$ vs. $D$ analysis (corroborated by the high $R^2$ in Figure 3(a)) strongly justify prior-

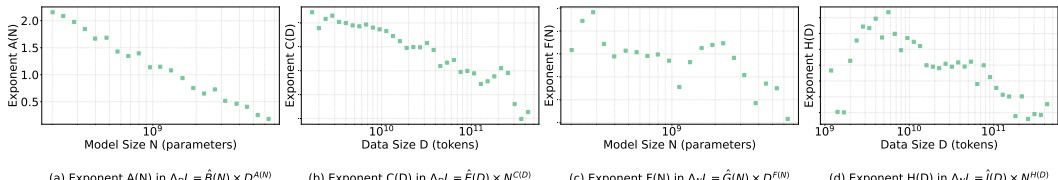

Figure 11: Log-Log projections of the exponent functions derived from four differential loss perspectives: (a) $A(N)$ in $\Delta_D L = \hat{B}(N)\, D^{-A(N)}$ vs. $N$; (b) $C(D)$ in $\Delta_D L = \hat{E}(D)\, N^{-C(D)}$ vs. $D$; (c) $F(N)$ in $\Delta_N L = \hat{G}(N)\, D^{-F(N)}$ vs. $N$; (d) $H(D)$ in $\Delta_N L = \hat{I}(D)\, N^{-H(D)}$ vs. $D$. Perspective (a), $A(N)$ vs. $N$, exhibits the most stable trend, reinforcing the choice of modeling $\Delta_D L$ as a function of $D$.

itizing this specific differential view. Thus, we adopt the form $\Delta_D L(N, D) \approx B(N) D^{-A(N)}$ as the basis for modeling the data-dependent terms.

**C.3 Functional Form Selection for $A(N)$ and $B(N)$**

Once the discrete sets of parameters $\{A_N\}$ and $\{B_N\}$ are obtained for each model size $N$ (Stage 1), continuous functions $f_A(N; \theta_A)$ and $f_B(N; \theta_B)$ are determined (Stage 2). This selection is guided by a systematic search over a dictionary of simple, monotone transformations $\mathcal{G} = \{\text{identity, logarithm, power functions}\}$.

For every ordered quadruple of transformations $(g_i, g_j, g_k, g_m) \in \mathcal{G}^4$, we perform coordinate projections:

$$g_i(A_N) = a_1 \, g_j(N) + b_1 + \varepsilon_{A,N} \tag{17}$$

$$g_k(B_N) = a_2 \, g_m(N) + b_2 + \varepsilon_{B,N} \tag{18}$$

The coefficients $(a_1, b_1)$ and $(a_2, b_2)$ are found via least squares for each combination. The transform quadruple $(g_i^*, g_j^*, g_k^*, g_m^*)$ that minimizes the sum of projection-space residual sums of squares, $\ell_A + \ell_B = \sum_N \varepsilon_{A,N}^2 + \sum_N \varepsilon_{B,N}^2$, is selected.

As stated in Section 3.2, empirical results indicated that applying a logarithmic transformation to $A_N$ and $B_N$, and a power-law transformation to $N$, yielded the best fit. That is, $g_i^*(A_N) = \log(A_N)$, $g_j^*(N) = N^\alpha$, $g_k^*(B_N) = \log(B_N)$, and $g_m^*(N) = N^\beta$. These choices lead to the stretched-exponential forms:

$$f_A(N; \theta_A) = \exp(a_1 N^\alpha + b_1) \tag{19}$$

$$f_B(N; \theta_B) = \exp(a_2 N^\beta + b_2) \tag{20}$$

The exponents $\alpha$ and $\beta$ are subsequently refined via an iterative strategy to minimize the global error of the loss differences $\ell_R = \sum_N \sum_D (R_N(D) - \tilde{R}_N(D))^2$.

A wide range of functional form combinations from $\mathcal{G}^4$ was evaluated. Table 2 presents a comparison of the top-performing candidates, thereby illustrating the types of transformations considered and highlighting the combination that empirically minimized the $\ell_A + \ell_B$ error.

Table 2: Summary of the functional form selection strategy for $A(N)$ and $B(N)$ using the transformation dictionary $\mathcal{G}$. The combination $(g_i = \log, g_j = \text{power}, g_k = \log, g_m = \text{power})$ was empirically found to minimize $\ell_A + \ell_B$, leading to stretched-exponential forms.

| $g_i(A_N)$ | $g_j(N)$ | $g_k(B_N)$ | $g_m(N)$ | Functional Form Type | Relative $\ell_A + \ell_B$ |
|---|---|---|---|---|---|
| Identity | Power | Identity | Power | Power-law + Power-law | 0.1322 |
| Log | Power | Log | Power | Stretched Exp + Stretched Exp | **0.1125** |
| Log | Power | Identity | Power | Stretched Exp + Power-law | 0.1148 |
| Identity | Power | Log | Power | Power-law + Stretched Exp | 0.1306 |

The quality of fit using these stretched exponential forms is demonstrated in Figure 4 in the main paper.

**C.4 Functional Form Selection for Model-Dependent Residual $E + L_N(N)$**

After fitting $L_D(D) + L_{ND}(N, D) \approx f_B(N; \theta_B^*) D^{-f_A(N; \theta_A^*)}$, the model-dependent residual term $E + L_N(N)$ is estimated. This is done by first computing $O(N, D) = L(N, D) - f_B(N; \theta_B^*) D^{-f_A(N; \theta_A^*)}$ and then averaging over $D$ to get $G(N) = \text{Avg}_D[O(N, D)] \approx E + L_N(N)$.

The functional form for $f_{L_N}(N; \theta_{L_N})$ (which models $G(N)$, absorbing $E$ as a constant within the fit) is determined by repeating a similar transformation-fit-selection procedure as for $A(N)$ and $B(N)$. For each pair of transformations $(g_p, g_q) \in \mathcal{G}^2$, we regress:

$$g_p(G(N)) = a_3 \, g_q(N) + b_3 + \varepsilon_{U,N} \tag{21}$$

The pair $(g_p^*, g_q^*)$ that minimizes the residual sum of squares $\ell_U = \sum_N \varepsilon_{U,N}^2$ is chosen.

As detailed in Section 3.3, the optimal transformations identified were a logarithmic transformation for $G(N)$ ($g_p^*(G(N)) = \log(G(N))$) and a power-law transformation for $N$ ($g_q^*(N) = N^\gamma$). This selection results in a stretched-exponential form for $E + L_N(N)$:

$$E + L_N(N) \simeq G(N) \approx \exp(a_3 N^\gamma + b_3) \tag{22}$$

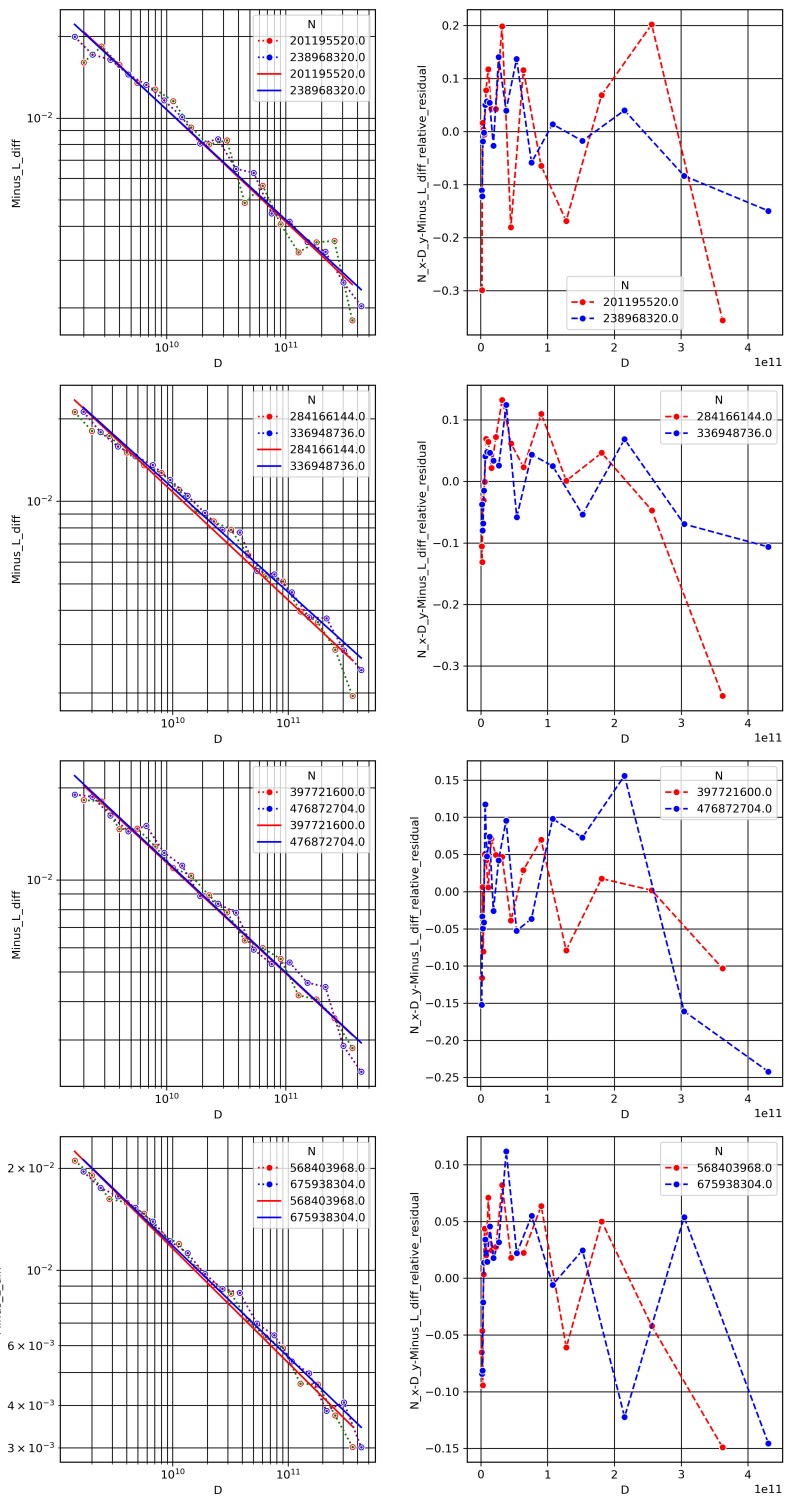

Figure 12: Example of the power-law relationship between the finite difference of the loss, denoted here as $Minus\_L\_diff$, and the data size $D$ for model sizes ($N$) from 201228288 to 676012032. (a) The left panel log-log plot shows a clear linear trend, indicative of a power law. (b) The right panel relative residuals of the fit are shown as a function of $D$, demonstrating the quality of the power-law approximation.

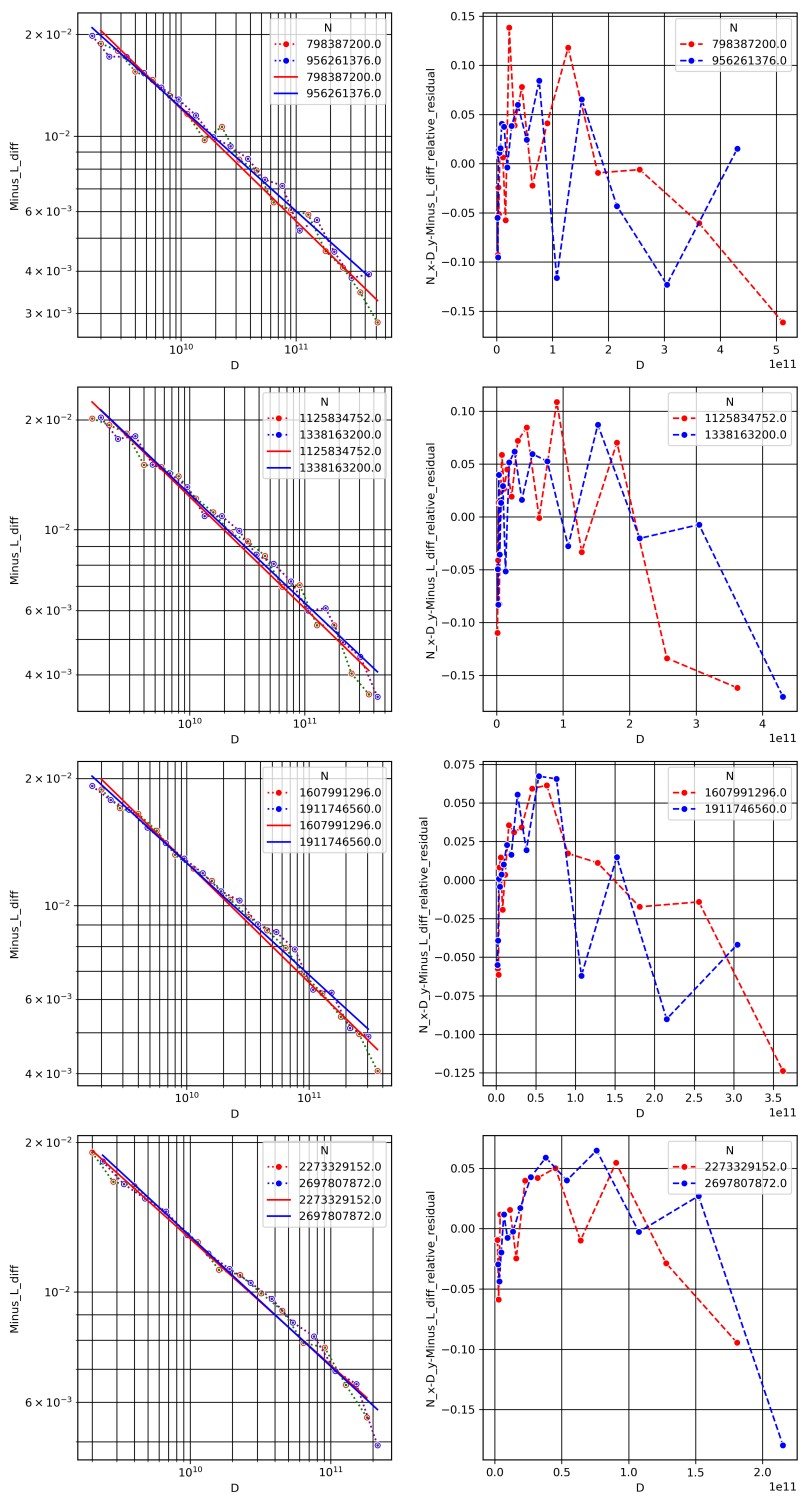

Figure 13: Example of the power-law relationship between the finite difference of the loss, denoted here as $Minus\_L\_diff$, and the data size $D$ for model sizes ($N$) from 798470400 to 2697992704. (a) The left panel log-log plot shows a clear linear trend, indicative of a power law. (b) The right panel relative residuals of the fit are shown as a function of $D$, demonstrating the quality of the power-law approximation.

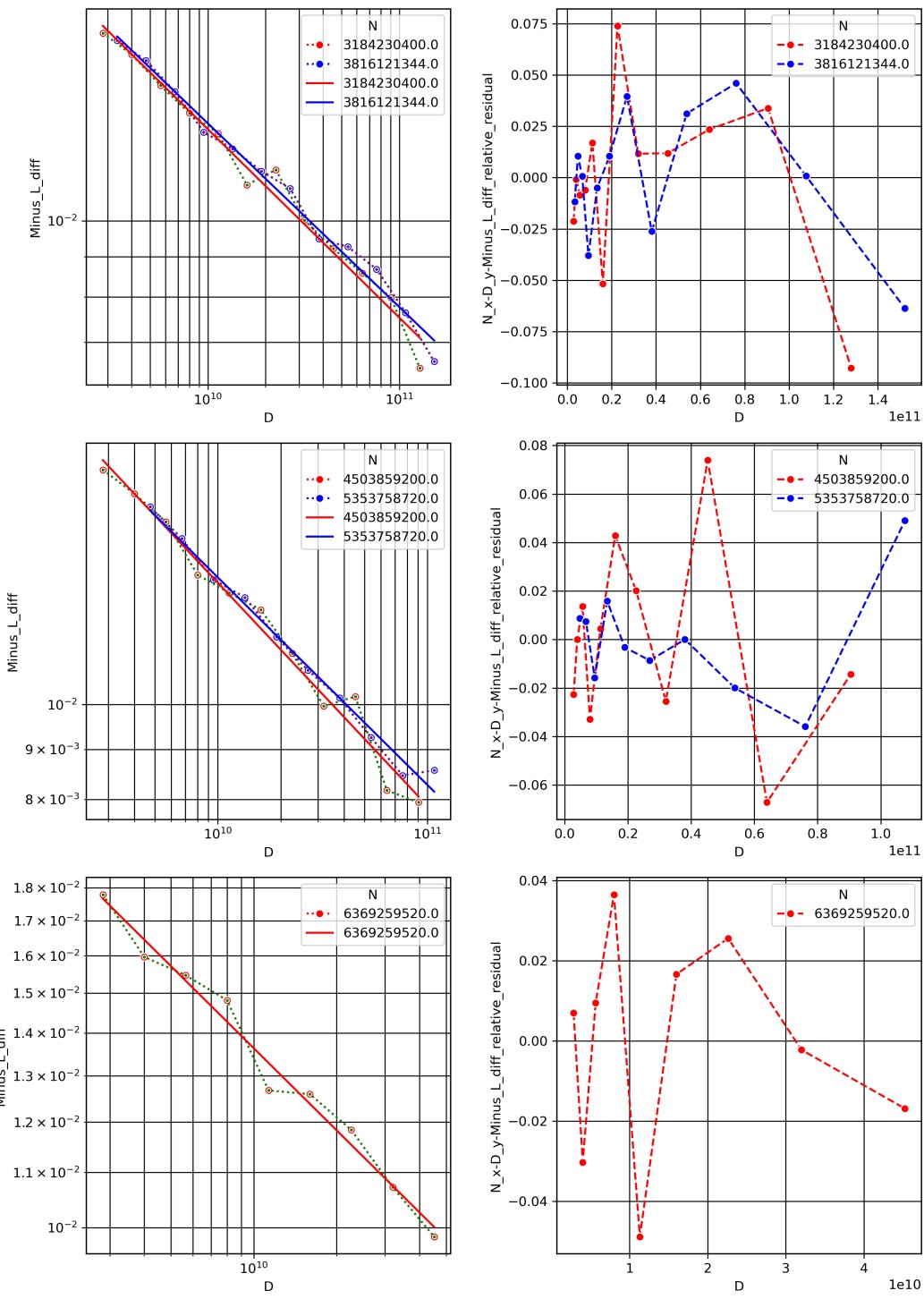

Figure 14: Example of the power-law relationship between the finite difference of the loss, denoted here as $Minus\_L\_diff$, and the data size $D$ for model sizes ($N$) from 3184435200 to 6369572352. (a) The left panel log-log plot shows a clear linear trend, indicative of a power law. (b) The right panel relative residuals of the fit are shown as a function of $D$, demonstrating the quality of the power-law approximation.

The exponent $\gamma$ is identified via a grid search minimizing the fitting error.

Table 3 summarizes this selection process, illustrating the types of transformations explored and highlighting the empirically optimal choice. The "Minimized Error" column refers to the $\ell_U$ criterion.

Table 3: Summary of the functional form selection strategy for $G(N) \approx E + L_N(N)$ using the transformation dictionary $\mathcal{G}$. The combination $(g_p = \log, g_q = \text{power})$ was empirically found to minimize $\ell_U$, leading to a stretched-exponential form.

| $g_p(G(N))$ | $g_q(N)$ | Resulting Functional Form Type | Relative $\ell_U$ |
|---|---|---|---|
| Identity | Power | Power-law | 0.0083 |
| Log | Power | Stretched Exponential | **0.0011** |

## D  Validation Set and Compression Rate Metrics

### D.1  Validation Set

A validation set must be based on the following principles: (1) Unseen integrity: All validation samples are rigorously excluded from the training data of any model. (2) Bias-free composition: Strict prohibition of model-generated or model-processed content to prevent overfitting towards specific model families. (3) Diversity: Ensuring coverage in various fields through extensive collection and random sampling protocols. (4) High quality: Validation samples must maintain strict syntactic correctness and logical integrity to avoid distortion of measurements. We construct a specialized English validation set comprising web pages, books, and academic papers, with each data category containing approximately 10 million tokens. The web data originated from SERP (Search Engine Results Page) pages crawled via Google searches targeting Wikipedia entities. Book data was digitized from newly published textbooks through our proprietary PDF parser pipeline, while academic papers are sourced from arXiv publications after 2024. This composite dataset is systematically organized into 55 distinct domains spanning culture, arts, history, entertainment, and other fields. For each data category, we implemented a rigorous filtering pipeline: preliminary data pruning including removal of books/papers with character lengths <2,000 and webpages scoring below 0.6 via our defect detection model, domain-stratified downsampling, and GPT-4o-based quality scoring with subsequent selection of top-ranked samples. Through the multi-stage refinement process, we ultimately develop a high-quality validation set containing 30 million tokens.

### D.2  Compression Rate Metrics

We adopt bits per character (BPC) as the quantitative metric for evaluating LLM compression efficiency. BPC measures the cross-entropy between the model's predicted distribution and the true character distribution. Crucially, when the predicted distribution aligns perfectly with the true distribution, this cross-entropy formulation becomes mathematically equivalent to lossless compression. This equivalence establishes BPC as a theoretically grounded metric for quantifying how effectively LLMs compress given text in our validation set. Formally, BPC is defined as:

$$BPC = \sum_{i=1}^{N} \log\left(\frac{1}{p_i}\right) \cdot \frac{1}{M} = \underbrace{\frac{1}{N}\sum_{i=1}^{N}(-\log(p_i))}_{\text{train loss}} \cdot \frac{N}{M} = R_1 \cdot R_2 = R \tag{23}$$

where $M$, $N$ denote corpus size and vocabulary size, respectively. $R_1$, the first term of the BPC formula, captures model compression through next-token prediction loss, which is equivalent to the training loss upon first exposure. $R_2$, the second term of the BPC formula, represents vocabulary compression rate. It is worth noting that due to differences in vocabulary size, cross-family model comparisons need to consider $R_1 \cdot R_2$, while analysis within the same family can focus solely on $R_1$.

## E  Point, Line, and Surface Comparison

One of the key applications of *Farseer* is to enable systematic comparison across different settings—such as alternative model architectures or pre-training data mixtures—using only modest

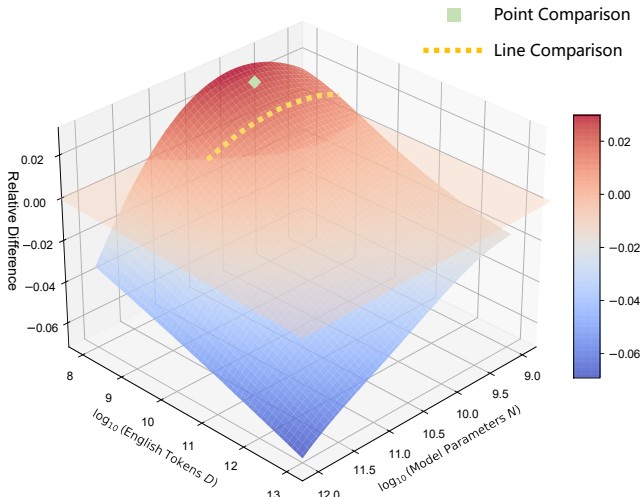

Figure 15: *Farseer*'s normalized 3D surface of relative BPC difference between datasets with 85% and 50% English proportions. The translucent pink plane marks $\Delta = 0$: above it, the 50%-English configuration outperforms; below it, the 85%-English configuration outperforms. Green squares show small-scale experiments at individual $(N, D)$ points, and the yellow dashed curve connects several such points—conclusions from these point/line comparisons do not hold at larger scales.

computational resources. By fitting two *Farseer* curves on relatively small-scale experiments, we obtain predicted loss surfaces $L(N, D)$ that faithfully capture performance across the full $(N, D)$ plane for each setting. This gives rise to a comprehensive *surface comparison* that subsumes and extends traditional point and line analyses: a **point comparison** highlights performance at individual $(N, D)$ coordinates; a **line comparison** reveals trends along fixed $N$ or $D$ slices; and the **surface comparison** synthesizes these insights into a continuous two-dimensional map of relative behavior.

To illustrate, we apply this to two pre-training mixtures with 85% versus 50% English data (balanced by Chinese for the remainder). We sample a modest grid of model sizes $N$ and English token counts $D$, fit *Farseer* to each subset, and compute the normalized relative BPC difference

$$\Delta = \frac{\mathrm{BPC}_{85\%} - \mathrm{BPC}_{50\%}}{\mathrm{BPC}_{50\%}}.$$

Fig. 15 shows the 3D surface of $\Delta$ over $\log_{10} N$ and $\log_{10} D$, with the zero plane marking parity between the two mixtures. Regions above the plane indicate that the 50% English mixture yields lower error, while regions below favor the 85% mixture. Green squares denote individual point comparison experiments at specific $(N, D)$ coordinates, and the yellow line connects several such points for a line comparison. Although these smaller scale analyses can suggest one mixture is better, the surface comparison reveals how those conclusions can reverse at larger scales. *Farseer*'s exhibits power for low-cost, high-fidelity extrapolation across any two training recipes or model designs.

## F   Model Parameter Configuration Table

Our model configurations are built upon the architectural principles of the LLama [39, 40], scaling depth, width, and attention mechanisms in a systematic grid to explore performance across a broad spectrum of model sizes. Table 4 provides an overview of the 21 distinct configurations (Model 1 through Model 21) used to fit the *Farseer* with Baseline training data in Tab. B. Each model (from the most compact Model 1 to the most expansive Model 21) was trained on an exceptionally large and diverse corpus, ensuring robust generalization and comprehensive coverage of the underlying physical phenomena:

- **Parameter scale:** The second column $N$ reports the number of trainable parameters outside of the embedding layers, ranging from $2.01 \times 10^8$ up to $6.37 \times 10^9$. This systematic scaling of model size enables us to rigorously examine how parameter count $N$ influences the behavior captured by *Farseer* for modeling scaling laws.

Table 4: Model configurations used in the *Farseer* study with Baseline data recipe, detailing architectural and training dataset properties across 21 systematically scaled transformer models.

| Model | $N$ | $D_{\text{count}}$ | $D_{\text{range}}$ | $d_{\text{model}}$ | $d_{\text{ff}}$ | $N_{\text{head}}$ | $N_{\text{layer}}$ | $N_{\text{with\_emb}}$ |
|---|---|---|---|---|---|---|---|---|
| 1 | 201228288 | 17 | $[1.41 \times 10^9, 3.62 \times 10^{11}]$ | 1024 | 2728 | 16 | 16 | 335446016 |
| 2 | 239005312 | 18 | $[1.19 \times 10^9, 4.31 \times 10^{11}]$ | 1088 | 2856 | 17 | 17 | 381611648 |
| 3 | 284207616 | 18 | $[1.00 \times 10^9, 3.62 \times 10^{11}]$ | 1152 | 3032 | 18 | 18 | 435202560 |
| 4 | 336994944 | 18 | $[1.19 \times 10^9, 4.31 \times 10^{11}]$ | 1216 | 3240 | 19 | 19 | 496378496 |
| 5 | 397772800 | 17 | $[1.41 \times 10^9, 3.62 \times 10^{11}]$ | 1280 | 3472 | 20 | 20 | 565544960 |
| 6 | 476931840 | 18 | $[1.19 \times 10^9, 4.31 \times 10^{11}]$ | 1344 | 3584 | 21 | 22 | 653092608 |
| 7 | 568468736 | 18 | $[1.00 \times 10^9, 3.62 \times 10^{11}]$ | 1472 | 3888 | 23 | 22 | 761406720 |
| 8 | 676012032 | 18 | $[1.19 \times 10^9, 4.31 \times 10^{11}]$ | 1536 | 4064 | 24 | 24 | 877338624 |
| 9 | 798470400 | 18 | $[1.41 \times 10^9, 5.12 \times 10^{11}]$ | 1600 | 4264 | 25 | 26 | 1008185600 |
| 10 | 956354688 | 18 | $[1.19 \times 10^9, 4.31 \times 10^{11}]$ | 1728 | 4528 | 27 | 27 | 1182847104 |
| 11 | 1125938688 | 18 | $[1.00 \times 10^9, 3.62 \times 10^{11}]$ | 1792 | 4832 | 28 | 29 | 1360819712 |
| 12 | 1338278400 | 18 | $[1.19 \times 10^9, 4.31 \times 10^{11}]$ | 1920 | 5184 | 30 | 30 | 1589936640 |
| 13 | 1608122368 | 17 | $[1.41 \times 10^9, 3.62 \times 10^{11}]$ | 2048 | 5448 | 32 | 32 | 1876557824 |
| 14 | 1911894528 | 17 | $[1.19 \times 10^9, 3.04 \times 10^{11}]$ | 2176 | 5712 | 34 | 34 | 2197107200 |
| 15 | 2273495040 | 14 | $[1.41 \times 10^9, 1.81 \times 10^{11}]$ | 2304 | 6064 | 36 | 36 | 2575484928 |
| 16 | 2697992704 | 15 | $[1.68 \times 10^9, 2.15 \times 10^{11}]$ | 2432 | 6488 | 38 | 38 | 3016759808 |
| 17 | 3184435200 | 13 | $[2.00 \times 10^9, 1.28 \times 10^{11}]$ | 2560 | 6952 | 40 | 40 | 3519979520 |
| 18 | 3816352512 | 13 | $[2.38 \times 10^9, 1.52 \times 10^{11}]$ | 2752 | 7336 | 43 | 42 | 4177062656 |
| 19 | 4504118400 | 12 | $[2.00 \times 10^9, 9.05 \times 10^{10}]$ | 2880 | 7744 | 45 | 45 | 4881605760 |
| 20 | 5354047488 | 11 | $[3.36 \times 10^9, 1.08 \times 10^{11}]$ | 3072 | 8264 | 48 | 47 | 5756700672 |
| 21 | 6369572352 | 11 | $[2.00 \times 10^9, 9.05 \times 10^{10}]$ | 3328 | 9136 | 52 | 47 | 6805779968 |

- **Data coverage:** Column $D_{\text{count}}$ indicates the number of distinct data subsets incorporated into each model's training process. The corresponding column $D_{\text{range}}$ (expressed in scientific notation) reports the minimum and maximum token counts used for training each model. Across all models, the training corpus spans roughly $1.00 \times 10^9$ to $5.12 \times 10^{11}$ tokens, highlighting the substantial scale and heterogeneity of the data leveraged to capture *Farseer*'s complex $D$ scaling behavior.

- **Architectural details:** Following the LLama design [39, 40], each configuration specifies the transformer hidden dimension $d_{\text{model}}$, feed-forward sublayer size $d_{\text{ff}}$, number of attention heads $N_{\text{head}}$, number of transformer layers $N_{\text{layer}}$, and total embedding parameters $N_{\text{with\_emb}}$. This structured sweep follows common design principles established by existing models, helping ensure that *Farseer*'s results remain compatible and comparable with prior scaling studies.

- **Data-rich training:** By combining extensive token budgets with carefully designed models, we ensure that even our largest architectures remain data-saturated, avoiding under-fitting in any scale regime. This abundance of training examples is critical for capturing the multi-scale interactions inherent in the *Farseer* phenomena.

This systematic sweep of model scales, grounded in widely adopted architectural designs and supported by substantial and diverse training data, provides a consistent and robust foundation for analyzing the scaling behavior captured by *Farseer*.

Additionally, Section 4.2 investigates the impact of different dataset compositions on *Farseer*. Under the EN-ZH configuration in Tab. B, we conducted experiments using the model settings presented in Table 5 to demonstrate the data generalization capabilities of *Farseer*, with settings analogous to those described above.

Table 5: Model configurations used in the *Farseer* study with EN-ZH data recipe, detailing architectural and training dataset properties across 9 systematically scaled transformer models.

| Model | $N$ | $D_{\text{count}}$ | $D_{\text{range}}$ | $d_{\text{model}}$ | $d_{\text{ff}}$ | $N_{\text{head}}$ | $N_{\text{layer}}$ | $N_{\text{with\_emb}}$ |
|---|---|---|---|---|---|---|---|---|
| 1 | 201228288 | 13 | $[2.00 \times 10^9, 1.28 \times 10^{11}]$ | 1024 | 2728 | 16 | 16 | 335446016 |
| 2 | 284207616 | 13 | $[2.00 \times 10^9, 1.28 \times 10^{11}]$ | 1152 | 3032 | 18 | 18 | 435202560 |
| 3 | 397772800 | 13 | $[2.00 \times 10^9, 1.28 \times 10^{11}]$ | 1280 | 3472 | 20 | 20 | 565544960 |
| 4 | 585641088 | 13 | $[2.00 \times 10^9, 1.28 \times 10^{11}]$ | 1472 | 3888 | 22 | 23 | 778579072 |
| 5 | 778000000 | 13 | $[2.00 \times 10^9, 1.28 \times 10^{11}]$ | 1600 | 4264 | 26 | 25 | 987715200 |
| 6 | 1099958272 | 13 | $[2.00 \times 10^9, 1.28 \times 10^{11}]$ | 1792 | 4832 | 29 | 28 | 1334839296 |
| 7 | 1608122368 | 13 | $[2.00 \times 10^9, 1.28 \times 10^{11}]$ | 2048 | 5448 | 32 | 32 | 1876557824 |
| 8 | 2273495040 | 13 | $[2.00 \times 10^9, 1.28 \times 10^{11}]$ | 2304 | 6064 | 36 | 36 | 2575484928 |
| 9 | 3184435200 | 13 | $[2.00 \times 10^9, 1.28 \times 10^{11}]$ | 2560 | 6952 | 40 | 40 | 3519979520 |

# G    Consideration of Parameters with Embedding Layers

A persistent question in the formulation and analysis of scaling laws for language models revolves around the inclusion of embedding layer parameters when defining the total model size, denoted as $N$. This is not a trivial accounting detail, as the embedding layer can constitute a significant fraction of a model's parameters, particularly for models with large vocabularies. Different perspectives on this issue have been adopted in seminal works. For instance, the scaling laws proposed by OpenAI often focused on non-embedding parameters, whereas the Chinchilla scaling laws by DeepMind typically accounted for the total number of parameters, including embeddings. This divergence in methodology highlights an underlying uncertainty regarding the effective contribution of embedding parameters to model capacity as characterized by scaling laws.

To investigate this ambiguity and determine a more empirically grounded approach, we conducted a targeted ablation study. The study aimed to directly compare the predictive power of scaling laws fitted using two distinct definitions of $N$: one that includes the parameters of the embedding layer and another that excludes them. We fitted separate scaling law functions to our experimental data under these two conditions.

The findings, as illustrated in Fig 16. In the vicinity of the data points used for fitting the scaling laws, both formulations – $N$ with and without embedding parameters – yield comparable predictions, exhibiting good local fits. However, a crucial distinction emerges when extrapolating to model sizes significantly larger than those in the fitting set. We validated the fitted laws at a 25.1 billion parameter scale (denoted as $N_{val} = 25.1B$). At this larger validation point, the discrepancy between the two an N with embedding layers scaled law, the error in prediction increased substantially to

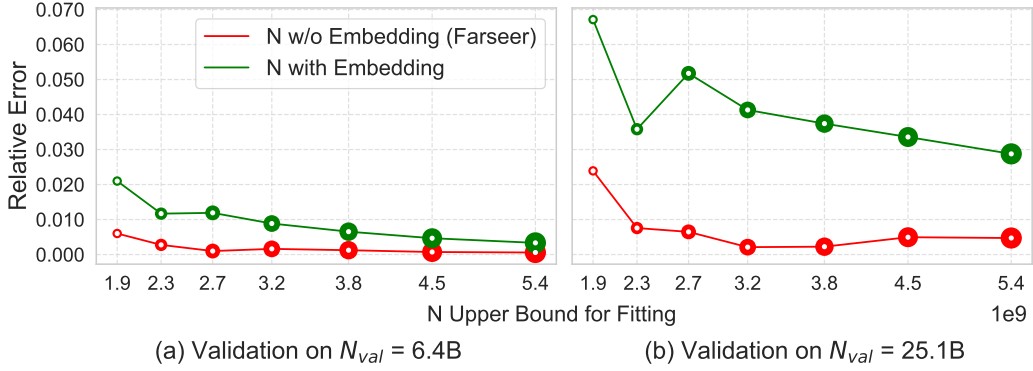

(a) Validation on $N_{val}$ = 6.4B          (b) Validation on $N_{val}$ = 25.1B

Figure 16: Scaling law extrapolation with and without counting embedding parameters in model size $N$. While both versions fit the training range well, only the formulation excluding embeddings extrapolates accurately to $N_{val} = 25.1B$. This indicates that embedding parameters may not scale with model capacity in the same way as transformer parameters, and excluding them yield more predictive scaling laws.

0.029. In contrast, the scaling law derived using N exclusive of embedding parameters demonstrated significantly better extrapolation, with an error more than four times smaller than its counterpart.

This ablation study provides compelling evidence that, from the perspective of achieving robust and generalizable scaling laws, it is preferable to exclude embedding layer parameters from the calculation of $N$. The improved extrapolation accuracy observed when omitting these parameters suggests that they may not contribute to the model's scalable capacity in the same manner as the core "compute" parameters of the transformer architecture. This finding has important implications for accurately predicting the performance of much larger models and for optimally allocating parameter budgets in future language model development.

# H   Non-linear End-to-End Fitting

To ensure a robust and equitable comparison of the predictive capabilities of the Farseer and Chinchilla scaling laws, we meticulously evaluated two fitting methodologies for both. This appendix details the procedures undertaken and the rationale behind the specific fitting choices reported in the main text.

Initially, our Differential Piecewise Fitting method (as detailed in Section 3 and Appendix C) was applied to both the Farseer and Chinchilla formulations using our comprehensive dataset (Appendix B). This allowed for a direct comparison of the models when subjected to the same fitting paradigm.

Subsequently, we also employed standard end-to-end non-linear regression for both models, mirroring the conventional approach used for Chinchilla. For the Farseer model, this process was particularly intensive due to its more complex functional form. To thoroughly explore the parameter space and mitigate issues with local minima, we conducted an extensive search involving over 30,000 optimization runs, each initiated with different prior-informed parameter initialization.

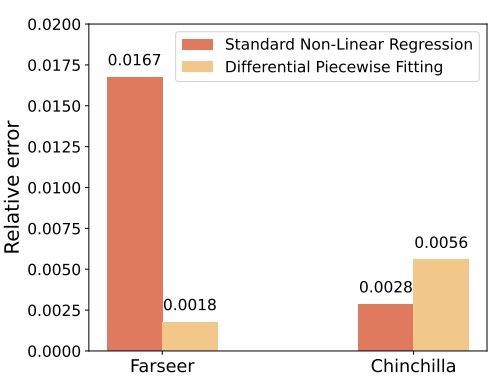
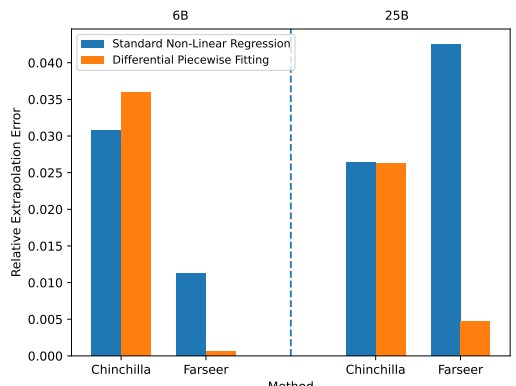

(a) Relative errors of Chinchilla and Farseer models fitted with standard non-linear regression and differential piecewise fitting.

(b) Relative extrapolation error at 6B and 25B for Chinchilla and Farseer, comparing standard end-to-end regression v.s. Differential Piecewise fitting.

Figure 17: (a) Chinchilla performs slightly better with standard regression, while Farseer shows a significant gain from piecewise fitting. (b) For the Farseer, differential piecewise fitting reduces the extrapolation error by an order of magnitude.

Our evaluations revealed distinct behaviors for the two models under these fitting regimes. For the Chinchilla model, the standard non-linear regression yielded a marginally better fit compared to our method, as illustrated by Fig 17.

In contrast, the Farseer model exhibited significantly different results. Due to its inherent complexity, the standard non-linear regression approach struggled to converge to an optimal global solution, even with the extensive initialization strategy. The resulting fit produced errors approximately an order of magnitude (10 times) higher than those achieved with our method. The superior performance of the Differential Piecewise Fitting method for Farseer suggests that its piecewise or iterative refinement is better suited to navigate the intricate loss landscape of our formulation, indirectly validating the structural accuracy of the Farseer model itself.

Given these findings, and to ensure each model was represented by its most effective and accurately fitted version, the comparisons presented in the main body of this paper are based on:

The Chinchilla model fitted using standard non-linear regression. The Farseer model fitted using our Differential Piecewise Fitting method. This approach allows for a fair comparison by leveraging the fitting technique that best captures the predictive power of each respective scaling law according to our empirical evaluations.

# I Analysis of Formula Properties

As defined by the fitted form

$$L(N, D) = e^{-0.021 \cdot N^{0.169} - 0.091} + e^{88.01 \cdot N^{-0.1} - 6.287} \cdot D^{-e^{-0.124 \cdot N^{0.123} + 0.424}} \tag{24}$$

Fig. 18 exhibited the trend of BPC $L(N, D)$ with model size $N$ and data size $D$. *Farseer* exhibits a fundamental property: it strictly decreases with both $N$ and $D$. This guarantees that additional compute or data never degrades performance. This feature makes *Farseer* both intuitively and formally well-behaved as a large-scale scaling law.

Starting from the general form in Eq. (4) and other mentioned definitions, we group terms as

$$L(N, D) = L_N(N) + B(N) D^{-A(N)},$$

with

$$L_N(N) = e^{-0.021 \cdot N^{0.169} - 0.091}, \quad B(N) = e^{88.01 \cdot N^{-0.1} - 6.287}, \quad A(N) = e^{-0.124 \cdot N^{0.123} + 0.424}.$$

Each of these terms is strictly positive for all admissible values of $N$ and $D$.

**1. Partial derivative w.r.t. $D$.**

Differentiating with respect to $D$, we obtain:

$$\frac{\partial L}{\partial D} = \frac{\partial}{\partial D} \left[ L_N(N) + B(N) D^{-A(N)} \right] = -A(N) B(N) D^{-A(N)-1}.$$

This expression is strictly negative since all multiplicative components—$A(N)$, $B(N)$, and $D^{-A(N)-1}$—are positive for admissible values of $N$ and $D$. Consequently, the loss $L(N, D)$ monotonically decreases with increasing data size $D$, aligning with the intuitive expectation that more data improves generalization. This behavior is visualized in Fig. 19, which shows the surface of $\partial L/\partial D$ over a wide range of $(\log N, \log D)$.

**2. Partial derivative w.r.t. $N$.**

By the product and chain rules,

$$\frac{\partial L}{\partial N} = L_N'(N) + B'(N) D^{-A(N)} + B(N) \frac{d}{dN} \big( D^{-A(N)} \big).$$

Because the analytical form of $\partial L/\partial N$ is prohibitively complex, we resort to numerical verification to determine its sign. As illustrated in Fig. 20, the numerically computed $\partial L/\partial N$ remains negative across the entire range of $N$. Therefore, $L$ is confirmed to decrease monotonically as $N$ increases.

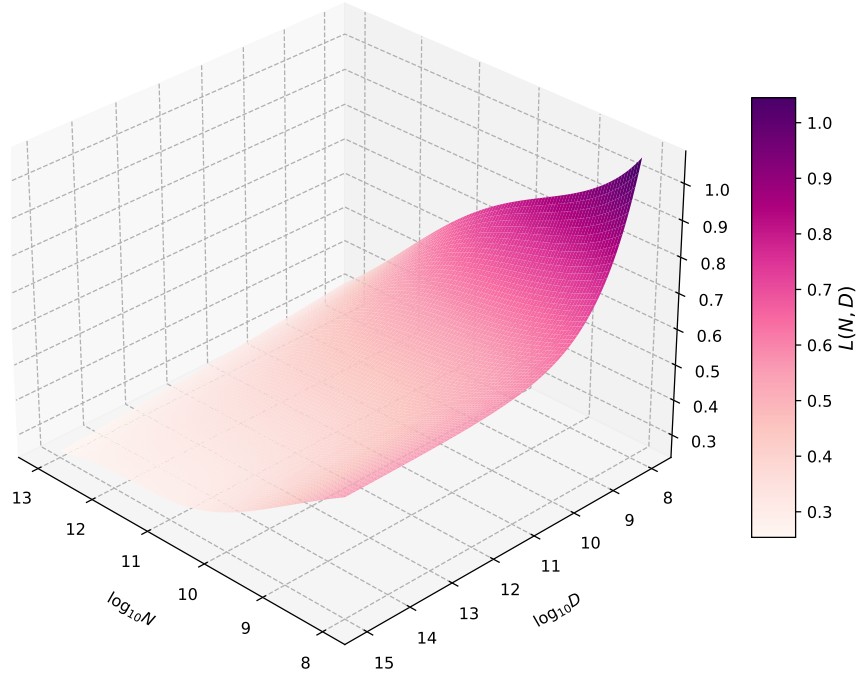

Figure 18: Visualization of $L(N, D)$ as a function of model size $N$ and data size $D$.

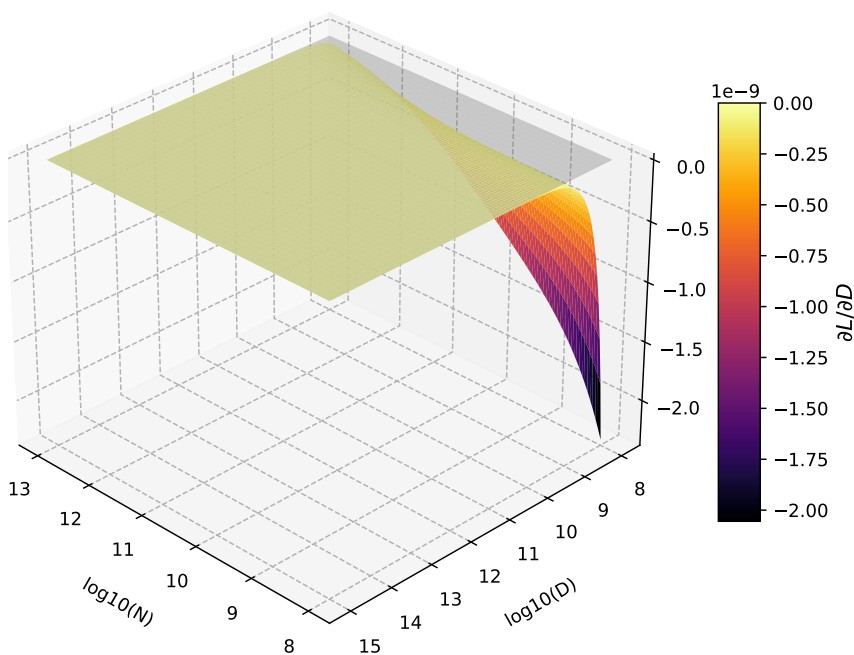

Figure 19: Visualization of $\partial L / \partial D$ as a function of model size $N$ and data size $D$, showing a consistently negative gradient across the domain.

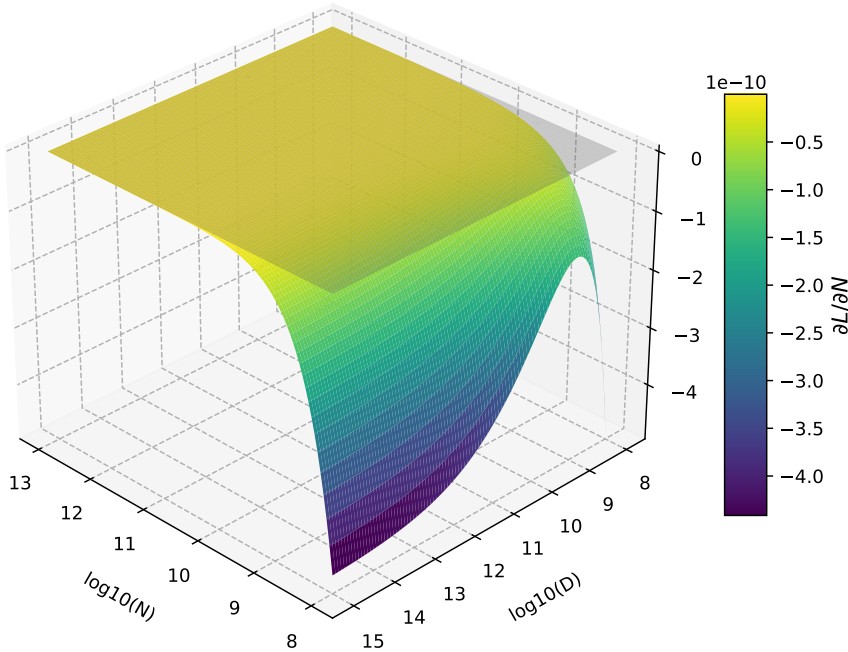

Figure 20: Visualization of $\partial L/\partial N$ over varying $N$ and $D$, also showing a consistently negative gradient across the domain.

## J   Discussion and Future Work

Through a rigorous experimental design, a massive number of experiments, and extensive experimental data collected over 18 months of experimentation and analysis, we have discovered a law that is more fundamental than the Chinchilla law, which we name the *Farseer*. On one hand, we believe this paper opens up a new field with a vast amount of potential work, such as deeper explorations of the Law based on Farseer, or comprehensive comparisons of various model architectures and data methodologies using Farseer. On the other hand, Farseer has some limitations. We will now delve into a detailed discussion of these two aspects.

**Computational power limitations.** Due to computational constraints, our evaluations have been restricted to Llama-style, decoder-only Transformers. Different model architecture paradigm's distinct coupling and optimization dynamics may alter the scaling coefficients and predictive accuracy. We have already begun experiments on MoE-based LLMs. Furthermore, due to computational limitations, this work only evaluates the applicability on two pre-training data distributions (an English-dominant distribution and a Chinese-English bilingual one). As this research began in early 2024, there was less focus on code data than there is now, hence our primary consideration of Chinese and English. However, we are currently investigating its applicability on code/math-dominant data distributions, validating on a code validation set. Please follow our subsequent work for updates. We are confident that the Farseer Scaling Law is applicable across different data distributions.

**Larger-scale validation.** This paper provides validation points up to a 25 B parameter LLM but does not include experiments on larger models, primarily due to computational power limitations. Other reasons include: (a) Farseer has maintained consistent prediction accuracy without any trend of increasing error across various extrapolation points of different sizes (3B/6.5B/25B). Therefore, there is reason to believe it can be generalized to larger models. (b) Larger models inherently have significant engineering implementation differences compared to smaller models. For instance, our models below 6.5B used only Data-Parallel distributed configurations, whereas for the 25B model training, we employed Pipeline Parallelism (Rank = 8) and Virtual Pipeline Parallelism (Rank = 5). Validating even larger models would require adopting further distributed training techniques, such as Tensor Parallelism. These distributed techniques themselves can introduce noise into the model's final performance. Therefore, for larger validation points (e.g., 135B), it would be necessary to run multiple different engineering implementations to reduce this noise, which poses further challenges

to computational resources. Balancing these complexities and benefits led to our decision to forgo larger-scale validations. (c) In numerous previous studies, a 25 B dense model is already past the threshold where various intelligent emergent phenomena appear. We believe this size is sufficient to support the adequacy of our extrapolation validation.

**BPC & downstream tasks.** We chose to use Bits Per Character (BPC) instead of specific tasks for evaluation based on the following reasons: (a) Compression rates measured on a large and diverse validation set reflect a model's general intelligence level, aligning with our goal of evaluating overall intelligence rather than task-specific performance. (b) As part of our experimental rigor, we meticulously constructed a validation set characterized by four properties: Unseen Integrity, Bias-free Composition, Diversity, and High Quality. Compared to individual benchmarks, it offers a more comprehensive assessment of intelligence and is guaranteed to be isolated from the training set. (c) Specific downstream tasks are highly susceptible to contamination and overfitting from training sets. In terms of magnitude, these benchmarks are more easily overfitted as evaluation targets. Consequently, pre-training datasets with varying degrees of contamination would exhibit different scaling laws, rendering such laws non-universal. (d) BPC possesses deep physical meaning related directly to compressibility, and numerous studies correlate a model's compressibility strongly with downstream task performance. Although our carefully constructed validation set has numerous advantages over downstream task benchmarks, with the continuous advancement of post-training techniques, whether a base model with a low BPC on the validation set necessarily implies a good model remains an open question. In summary, using BPC testing on a meticulously constructed validation set is the best method available to us at the current level of public technology. Its limitations are primarily constrained by the state of contemporary technology.

**Theoretical explanation.** This work is a characterization of scaling laws based on massive observational experimental data, and it does not provide a theoretical explanation. This is similar to all other work in the field of scaling laws, none of which have derived the specific formula of a scaling law from fundamental machine learning theory. This limitation, common across scaling law research, remains a significant open challenge in machine learning theory.

**Broader generalization.** Our research primarily investigates decoder-only Transformers optimized via Next Token Prediction. Generalizing *Farseer*-like formulations to broader machine learning tasks—such as multimodal or diffusion-based models—remains unexplored in this work.

**Open source.** We believe that our experimental data can be used for much further analysis, interpretation, or discovery of new patterns. Therefore, we have decided to open source our massive experimental data, worth millions of dollars, including 1000 LLM models trained from scratch and their entire training processes. We may even open source the training dataset and the meticulously constructed validation set. Based on this, We anticipate further research in at least the following directions: 1) **Formula and fitting.** We believe there may exist a more concise scaling law or a better fitting method. 2) **More applications.** While compute-optimal guidance is valuable for model training, the entire $L(N, D)$ surface obtained from Farseer certainly holds more valuable conclusions to be mined, such as the analysis of first-order partial derivatives, etc. 3) **Training dynamics.** This paper only evaluates the final performance of the models. The training dynamics of 1000 models of different model sizes and token sizes could be further studied to reveal underlying patterns.

This paper pioneers a new experimental methodology for the era of large models. Consider two experimental settings: base A and experimental B. Within each group, all settings are kept identical, including but not limited to LLM Architecture Family, Data Distribution, and Training Strategy. Under settings A and B, two sets of small models, Models A and Models B, are trained. Using the method provided in this paper, two bivariate functions, $L_A(N, D)$ and $L_B(N, D)$, are fitted. These two functions share the same functional form (Farseer) but have different parameters. We can then compare the superiority of the two settings under infinite and arbitrary configurations of N and D. Under the previous methodology, if both Setting A and Setting B use a 3B parameter size and 100B tokens, and the model from Setting A outperforms the model from Setting B, it only proves that A is better than B at that specific N, D size. Unlike conventional methods limited to specific configurations (e.g., fixed at 3B parameters, 100B tokens), our method identifies precise conditions under which one setting outperforms another. This extensibility fundamentally reshapes key insights, examples including:

**Code/Math.** What is the performance of code and math data with different mixture ratios on the scaling law? Will the Code Scaling Law differ from the Nature Language Scaling Law in various

parameters, thus leading to completely different compute-optimal character or different first-order derivatives with respect to N and D? Does a good Code/Math model require an ultra-large model size like nature language models do? We have already completed a portion of the relevant experiments, and this research is ongoing and will be published in a subsequent paper.

**Sparse LLMs.** Do LLMs based on the MoE architecture [7] also follow the same functional form and predictive accuracy? And how should an "Architecture Family" be defined within the MoE architecture? We have already run some experiments, and this research is ongoing and will be published in a subsequent paper.

**Linear attention.** This paper uses Multi-Head Attention (MHA), a full attention paradigm. However, some studies suggest that certain types of Linear Attention-based LLMs perform better than full attention on smaller models [17]. The Farseer experimental methodology allows for a comprehensive comparison of the performance of different types of linear attention and full attention under various N and D configurations. This could answer the key question, such as whether linear attention is only suitable for a smaller D/N ratio. We look forward to the community building upon Farseer to complete this type of work.

**Model architecture.** There are many more innovations at the model architecture level that need to be compared from a comprehensive scaling law perspective, such as Loop Transformer, etc. From this viewpoint, the focus is on whether the relevant innovations can maintain performance as $N$ scales up and whether performance saturates quickly as $D$ increases.

**Data quality and repetition.** Using high-quality data and employing different data repetition strategies, can performance be maintained as N and D increase? From the perspective of scaling laws, what is the trade-off relationship with larger amounts of middle-quality data?

**Synthetic data.** What patterns does synthetic data exhibit from a scaling law perspective? Is Long CoT-like synthetic data more difficult to saturate with learning as D increases compared to general synthetic data?

