# OpenReview forum: "Predictable Scale (Part II) --- Farseer: A Refined Scaling Law in LLMs"
_NeurIPS.cc/2025/Conference — NeurIPS 2025 spotlight_

### Official Review · Reviewer_gWVt · 2025-07-01

**Clarity:** 3
**Significance:** 3
**Originality:** 3
**Rating:** 5
**Confidence:** 4

**Summary:**

The authors of this paper introduce Farseer, a set of scaling laws designed to describe the performance of language models. The scaling laws proposed by the authors are designed to extrapolate well beyond the setting where the number of parameters and the number of training tokens are comparable to each other. The authors compare to another popular formulation for scaling laws in this setting (Chinchilla) and demonstrate better results in the extreme ends of the scaling of available compute.

**Questions:**

- I would be grateful if the authors could let me know whether they have looked into applying their ideas in predicting downstream performance, as mentioned above.

**Ethical Concerns:**

["NO or VERY MINOR ethics concerns only"]

**Final Justification:**

As I mentioned in my original review, I believe Farseer to be a very solid work, with clear improvements upon scaling laws proposed in prior works. The responses of the authors to both mine and other Reviewers' comments further reinforce this belief, with additional resutls and clarifications. As such, I am keeping my score of "Accept" for this paper.

**Limitations:**

The authors have adequately addressed the limitations and the impact of their work.

**Paper Formatting Concerns:**

No major concerns.

**Quality:**

3

**Strengths And Weaknesses:**

Strengths

- Scaling laws are a very important topic and of great interest to the NeurIPS community. As model and dataset sizes increase, experimentation becomes harder. Being able to iterate on smaller scales and extrapolate on larger ones is a very important capability, which allows for easier improvements upon existing techniques.

- The paper is very easy to follow and very clear in writing. The method itself is also presented in a very clear manner.

- As a method, Farseer is a clear improvement upon prior work. While prior scaling laws ignore the interaction terms between the number of training tokens and model size, Farseer explicitly incorporates this term in the scaling law. While this could lead to difficulties in fitting the scaling law itself, the authors have been able to present a clear technique to circumvent this (namely, differential piecewise fitting and iterative improvements over the fit). The proposed method is well motivated, with several experiments to determine the form of each term in the equation (by justifying a power law behavior for the data dependent part $V(D) + H(N,D)$, and using that as a starting point to predict the form of the rest of the terms combined).

- The empirical performance of the proposed scaling law is also very good. The authors demonstrate low relative error of the predicted loss for the model, even on points that were not used to fit the scaling law (which lie above the model sizes used for fitting), thus demonstrating good generalization. Comparisons with Chinchilla also demonstrate a clear improvement over prior forms of scaling laws.

Weaknesses

- While not necessarily a weakness, I believe the paper would be further improved by adding more baselines for comparison if possible, as well as potentially examining the applicability of the approach of Farseer in predicting the downstream performance of models as well (instead of only the loss).

- Some minor comments:

  - Figures 2 and 7 should float to top or bottom of page if possible, for readability.

  - In line 130, there is a LaTeX error for $\hat{B}_N$.

  - The syntax at lines 146 - 147 is a bit weird.

  - In line 168, the first sentence should be bolded.

---

> ### Author Rebuttal · Authors · 2025-07-31
>
> We sincerely thank you for your thorough review and positive assessment of our work. We are very encouraged by your recognition of our paper's clarity, novelty, and significance. Your insightful suggestions are invaluable for improving our manuscript.
> Below are our responses to your specific points.
>
> ---
> **W1 & Q1: More baselines and downstream tasks**
>
> Thank you for these excellent suggestions.
>
> **Regarding the inclusion of more baselines,** we focused our comparison on the Chinchilla scaling law because it represents the most widely recognized and dominant formulation in this area. Many other proposed scaling laws are often modifications of the Chinchilla framework or are tailored for very specific, narrow tasks. Therefore, we believe that a direct and thorough comparison with Chinchilla provides the clearest demonstration of Farseer's advancement over the current state-of-the-art.
>
> **Regarding the prediction of downstream task performance,** we agree this is a very important research direction. We did not include it in this work because establishing a precise and stable scaling law for downstream benchmarks is a substantial research challenge in its own right, due to several sources of high variance in their evaluation.
>
> Specifically, we identify three main issues:
> 1. Stochasticity in Decoding: Most benchmarks rely on an LLM decoding process followed by an answer-matching script, a method that is inherently probabilistic. A model's score can fluctuate significantly across identical runs. While this can be partially mitigated by averaging over multiple decoding attempts (i.e., avg@N), it introduces measurement noise that complicates the modeling of a clean scaling signal.
> 2. Benchmark Sensitivity and Scope: The performance on any single benchmark can be volatile. This arises because (a) benchmarks often have a limited number of test items, making scores susceptible to high variance, and (b) they are extremely sensitive to the model's output formatting and its ability to follow instructions. A pretrained model's instruction-following capability can change drastically over very few training iterations depending on recent data exposure. This can cause large swings in benchmark scores that reflect shifts in instruction adherence rather than fundamental changes in the model's core capabilities. To obtain a more stable signal, one might need to cluster similar benchmarks and predict an aggregate metric, which adds its own layer of modeling complexity.
> 3. Confounding Effects of Fine-Tuning: To fairly compare the underlying capabilities of different pretrained models, one cannot simply test them directly. A rigorous evaluation would require first aligning each model with a standardized, lightweight instruction-following fine-tuning. This would normalize their instruction-following abilities to a similar level before evaluation. However, designing and constructing such a benchmark instruction-following dataset is a highly demanding task in itself.
> For these reasons, directly modeling downstream performance introduces significant confounding variables that would obscure the scaling dynamics related to pre-training. We believe that rigorously addressing these measurement challenges to isolate a clean scaling law for downstream tasks is a substantial endeavor that warrants a dedicated future study. We have added a discussion of this as a promising direction for future work in our revised manuscript.
>
> **W2: Minor comments on typos and formatting**
>
> Thank you very much for your meticulous reading and for pointing out these issues. We appreciate you catching the formatting suggestions for Figures 2 and 7, the LaTeX error in line 130, the syntax in lines 146-147, and the missing bolding in line 168. We have corrected all of these in the updated version of the paper.
>
> Once again, we thank you for your valuable feedback and strong support for our paper. We hope our responses have satisfactorily addressed your questions.

---

> > ### Comment · Reviewer_gWVt · 2025-08-04
> > **Re: Rebuttal**
> >
> > Thank you very much for the extremely detailed and informative responses to both mine and the other Reviewers' comments.
> >
> > I believe that Farseer is a strong paper, and as such I am keeping my recommendation of "Accept".

---

### Official Review · Reviewer_NtZX · 2025-07-01

**Clarity:** 2
**Significance:** 2
**Originality:** 3
**Rating:** 4
**Confidence:** 5

**Summary:**

The paper revisits the data and model size dependent scaling laws in the literature and improves them by adding an additional term that is both data- and model size-dependent. This way the scaling rate as data/model size increases differs according to the particular model size)data.

**Questions:**

Around line 96, why isn't H(N, S) included in model-dependent terms?

**Ethical Concerns:**

["NO or VERY MINOR ethics concerns only"]

**Final Justification:**

I thank the authors for the clarifications. The response sufficiently addresses my points. Since the authors said these will be reflected in the revised paper, I increase my score.

**Limitations:**

I don't think there is enough (or any) coverage of limitations. The proposed law brings some improvements over the Chinchilla laws but it's still a model that comes with limitations. It would be great to see a discussion on this

Another obvious drawback that deserves a discussion is the increased number of parameters in the law compared to Chinchilla. This would require more small-scale models to be trained.

**Paper Formatting Concerns:**

I haven't noticed any issues.

**Quality:**

3

**Strengths And Weaknesses:**

Strengths:

- Modeling the interrelations between the data and model size at scale is very important. The paper does a good job demonstrating the existing laws' inability to capture such relations and show empirical gains over them

Weaknesses:

- It's not clear how the scaling law form on line 92 changes to the one on line 97. The definition of "f" comes later but it's confusing to use it on line 97 without an explanation.

- Around line 96, why isn't H(N, S) included in model-dependent terms?

- The proposed law, Farseer, has 6 degrees of freedom as opposed Chinchilla's 5 degrees of freedom. While it makes sense that this additional parameter is needed to capture the dependency between the effect of N and D, it also increases the number of small-scale models needed to be trained to fit the scaling laws. I suggest the authors discuss this additional cost in the paper.

-Minor: some typos and grammatical errors: "H(N,D) models represents on line 70, "detailed in detailed in" on line 77. Also, footnote 1 of page 3 appears at page 4.

---

> ### Author Rebuttal · Authors · 2025-07-31
>
> Thank you for your efforts and contributions in reviewing the paper. Below are our responses to the relevant issues; we hope they will address your concerns and improve the quality of the manuscript.
>
> ---
>
> **W1: Clarity on the transition from the formula on line 92 to line 97**
>
> Thank you for pointing out this lack of clarity. You are correct that the transition was abrupt and lacked proper explanation. The formula on line 92 is the general, high-level form that we hypothesize. The formula on line 97 is the specific, empirically-derived functional form that resulted from our analysis in Section 3.
> Specifically, through our data exploration, we found that the general terms could be effectively approximated as follows:
> - The term V(D)+H(N,D) is approximated by $f_B(N;θ_{B∗})D^{f_A(N;θA∗)}$
> - The term E+U(N) is approximated by $f_U(N;θ_{U∗})$
> We improperly present the final form on line 97 before detailing the derivation and justification. This section will be revised in the new version of the manuscript to ensure a logical flow, in which introduce the general scaling law and then systematically explain how our empirical findings lead to the specific functional forms, defining all terms before they are used.
>
> **W2: Why H(N, D) is not included in model-dependent terms**
>
> This is an excellent and perceptive question.Our choice to group H(N, D) with the data-dependent term V(D) is motivated by both theoretical precedent and strong empirical evidence from our own analysis.
> On one hand, our formulation is inspired by the theoretical analysis in the Chinchilla paper(Appendix D.2.) [1], which we use as a foundational starting point. In their work, the total loss is decomposed into:
>
> $$
> L(N, D) = L(f^\star) + \left(L(f_N) - L(f^\star)\right) + \left(L(\bar{f}_{N,D}) - L(f_N)\right)
> $$
>
> 1. A term limited by the model size (N), which is $L(f_N) - L(f^\star)$
> 2. A term, which we denote as H, that is related to the data size (D) and the training dynamics, which is $L(\bar{f}_{N,D}) - L(f_N)$
>
> On the other hand, our decision to treat the term $$H(N, D)$$ as a power-law function of the data size, $D$, was fundamentally an empirical one, guided by the results of our analysis.  Our first-order difference analysis (Section 3.1) provided strong evidence for this modeling choice. Specifically, as shown in Figure 3a, the analysis of $$\Delta_D L(N,D)$$ versus $$D$$ revealed a remarkably stable and clear power-law relationship, evidenced by a high $$R^2$$ value of 0.9807. This empirical observation led us to model $$H(N,D)$$ as a power law with respect to $D$.  Crucially, we recognize that this power-law relationship is not independent of the model size, $N$. Our findings indicate that the parameters of this power law, most notably the exponent, are themselves a function of $N$. Our key contribution, therefore, is demonstrating that the complex behavior of the $$H(N,D)$$ term can be effectively captured by modeling it as a power law in $D$, whose characteristics are modulated by $N$. We will clarify this distinction in the revised manuscript.
>
>
> We admit that the original manuscript did not explain this motivation clearly. In the revised version, we will explicitly state that we adopt the loss decomposition from Chinchilla  and then clearly articulate that our primary contribution is demonstrating and modeling the previously overlooked dependence of the H term on N.
>
> **W3: Increased degrees of freedom, fitting cost**
>
> Regarding the additional parameter:
> Our parameters is intentionally introduced and is crucial for capturing the interaction effect between model size (N) and data size (D), which, as our results show, is a key factor that existing laws fail to model.
> Regarding the fitting cost:
> 1. Farseer achieves a comparable and evne more accurate with  a small number of data points.
> While more parameters can sometimes require more data, our Farseer law demonstrates greater efficiency in capturing the true scaling behavior. In fact, our Figure 8 already provides a preliminary illustration of this, showing that Farseer achieves a more accurate fit even with a small number of data points.
> 2. The error  of Chinchilla becomes divergent, while Farseer's remains stable.
> This is a key contribution of our work. It is the empirical demonstration that Chinchilla's formulation, which lacks the higher-order interaction term H(N, D), leads to divergent prediction errors, as shown in Figure 1a and Figure 2. In contrast, Farseer's error remains convergent and stable across the full range of N.
> 3. Farseer is closer to the true nature of Scaling Law, not just a simple accuracy-cost trade-off. The primary reason for Chinchilla's divergent error is that it extrapolates using a single, average rate of data scaling derived from its fitting set to predict performance for all other model sizes. This means that as the target model size N for prediction grows larger and deviates from the fitting range, Chinchilla's error is expected to increase and diverge. Farseer, however, is designed to capture this variance by modeling the interaction between N and D, leading to higher predictive confidence for larger models. This is critical because in most industrial and research applications, the goal of using scaling laws is precisely to forecast the performance of these larger, more resource-intensive models.
> From another perspective, Chinchilla does not inherently benefit from being fitted on a wider range of model sizes (N). Farseer, on the other hand, leverages the additional data points across N to build a more accurate model of how data scaling dynamics change with model size. Therefore, this is not a simple trade-off between prediction accuracy and fitting cost. Instead, it concerns the fundamental capability of the scaling law to generalize reliably to the large-scale regimes that are of greatest practical interest.
> 4 . A scaling law is only valuable if its prediction error is within a reasonable range. Here, we wish to add an observation from our industrial practice. When we need to use large-scale models to determine key conclusions, such as the effectiveness of different Attention structures, experiments are often conducted at the scale of tens of billions of parameters (e.g., >20B) and hundreds of billions of tokens. In this context, a performance difference between two approaches must typically exceed 0.6% for us to consider it a significant gap, while differences below this threshold are often attributed to noise or engineering implementation details. In our experience, the performance gap between two distinct, viable solutions rarely exceeds 3%. However, as shown in Figure 8, Chinchilla's prediction error for a 25B model already surpasses 2.5%. An error of this magnitude severely diminishes its practical value, as it can obscure the true performance differences between models and lead to incorrect conclusions.
>
> **About limitations:**
>
> We had previously included an extensive two-page discussion of "Discussion and Future Work" in Appendix J. However, we completely agree that a more thorough and centralized discussion of limitations is needed. Prompted by your suggestion, we recognize that this topic deserves a more prominent treatment in the main paper. To that end, we have added a dedicated "Limitations" section to the revised manuscript. In this new section, we will not only summarize the various aspects we originally mentioned in Appendix J, but we will also condense and integrate our points regarding the "fitting cost" of Farseer(as mentioned above).
> We agree that adding this section to the main body makes the paper stronger, and we appreciate you pushing us to improve its clarity and completeness.
>
> [1] [Training Compute-Optimal Large Language Models](https://arxiv.org/abs/2203.15556)

---

> > ### Comment · Reviewer_NtZX · 2025-08-01
> >
> > I thank the authors for the clarifications. The response sufficiently addresses my points. Since the authors said these will be reflected in the revised paper, I increase my score.

---

### Official Review · Reviewer_B5rQ · 2025-07-01

**Clarity:** 3
**Significance:** 4
**Originality:** 3
**Rating:** 6
**Confidence:** 4

**Summary:**

This work proposes a new functional form and method for fitting tends in the loss surface of pretrained language models, L(N, D). Unlike the common baseline “Chinchilla” approach, this work considers the higher-order interaction between #params (N) and #data (D). They train a very large sweep of models up to ~10^10 params and ~10^11 tokens, and observed finite differences in loss wrt D lead a paramerizing the power law of D with N. They then use a differential piecewise and iterative fitting technique. The result is an estimate of L(N,D) that: 1) achieves better error than Chinchilla for small and large models, 2) suggests an optimal token-param ratio more dependent on compute size, 3) is robust across both an English and English+Chinese pretraining mix, 4) extrapolates well, e.g. Farseer overtakes Chinchilla on predictions of a 25.1B param model when using at least a 2.3B param model for fitting (though with 1.9B param model as the largest fitting model Chinchilla performs slightly better). Finally note that the authors release all “models, data from ∼1,000 trained LLMs, 61 detailed logs, and the Farseer code.”

**Questions:**

- Will you release training code or just Farseer fitting code? Glancing at the supplement the code there does not seem to include training code.

- Really like figure 7! Might be helpful to put earlier to get a quick summary of all the experiments you ran. Early on I was wondering if you needed some kind of specialized set of experiments to fit your method, but seeing this figure I understand you just need a grid of models with the grid step determined by lambda.

- The curated validation set in appendix D sounds very interesting! I would love to know more. Will it be released? Is the 2024 data cutoff intended to be after the data cutoff of the train data? Does the English+Chinese data also have a val set like this?

- I like that the various scaling law functional forms are listed directly in the intro! A small but nice though would be to just say that all other params than N and D are fitted.

- figure 2: What are the N values here? fixed? Why show this specific slice?

- Line 20: LM progress “attributed to scaling laws”; does this mean the discovery and use of scaling laws has largely driven progress in the field or does this mean that improvements in scale which are described by scaling laws are the mechanism of most improvement. The latter seems like a much less contentious assertion.

- line 45: “the very exploration for superior scaling laws is itself severely hampered by a significant scaling gap.” ([pdf](zotero://open-pdf/library/items/8LZPGLQW?page=2)) This seems to suggest that the issue with the scaling gap is that it inhibits research on how to fit scaling trends. But the rest of the paragraph describes how the scaling gap causes problems with applying predictions from existing scaling law methods to model development. The latter makes sense to me, but I’m not sure I follow the quoted section. What exactly do you mean by the scaling gap?

- figure 4: saying what A(N) and B(N) represent in direct language (the model-size dependencies of the data scale exponent and coefficient) would help the caption a lot and make the figure self contained. Its also hard to understand what it means to have an “actual” values for this until you read the running text.

- Figure 5: likewise give some explanation in the caption of G(.) U(.) etc…

- figure 6: the x axis label on the figure is confusing. The caption description is better. Maybe try just: “Largest model size used in fitting”.

    - This figure seems redundant with figure 8. It’s more helpful to see it with the chinchilla results as baseline.

    - This seems like a somewhat modest gap in compute (from 1B to 5B, I’m assuming D is comparable?), do you have results farther on the left of the x axis?

- eq 4: consider maybe using more mnemonic names for the components, e.g., “L_N(N)” instead of “U(N)”

- line 86: I think this is supposed to point to Appendix F not A right?

- figure 3: the ticks names are too sparse (sometimes just 1 per axis). Also maybe mark what Y axis direction indicates faster improving loss? Also the caption is a bit unclear to me. What does it mean to “adobt this form in our main analysis”? Also A(.) and B(.) are reffed before definition here. Also is the r^2 here is over all the points not just per line?

- line 130: typo in latex hat

- The motivation for what is being developed in 3.2 is hard to follow. Maybe start clarifying that you are trying to fit the lines in Figure 3 (a) and that stage 1 and 2 are about fitting the F_a and F_b. This would be clearer if instead of \hatR_N(D) you used like `\hat\delta_DL(N,D)`

- line 162: I think this ref should be fig5 (b)?

**Ethical Concerns:**

["NO or VERY MINOR ethics concerns only"]

**Final Justification:**

The authors' response reassured me that they have a good plan for revising the clarity issues I raised. I don't have any other significant concerns remaining. I think the paper will be very impactful as it convincingly unseats a fundamental assumption about base language modeling scaling by showing that optimal tokens to parameter ratio is dependent on compute scale rather than being captured by the fixed "Chinchilla" rule of thumb. This finding should not just change practices in industry but will help clarify many scientific findings as understanding when an experiment is or is not overtrained is essential to reaching accurate conclusions to many questions.

**Limitations:**

The paper does not have a dedicated limitations section in the body of the paper but does discuss limitations in the appendix

**Quality:**

4

**Strengths And Weaknesses:**

Strengths:

1. This is a significant open source scientific release. To my knowledge this is the largest sweep over #params and #tokens, and will likely become a standard resource for future scaling law studies. They also release their data (hopefully both pretraining and validation) which is still relatively rare for works of this scale and can help all other kinds of research such as membership inference. And they also release their code, though the submission does not clarify if this includes training code or just loss surface fitting code.

2. I’m particularly impressed that they did a robustness test over a different pretraining data distribution!

3. Beyond having improved error over chinchilla scaling laws, their results also give us a more fundamental insight into optimal compute allocation, which raises an interesting question as to why optimal token/param ratio appears to increase with total compute budget. Similarly staring at their observed finite differences in loss in figure 3 is fascinating for starting to puzzle through the deeper dynamics here.

4. Provides a practical insight for developers that we should abandon the chinchila rule of thumb where optimal token-param ratio is often assumed as a constant 20.


Weaknesses:

1. It is difficult to judge how significant the improvement over existing methods is in practical terms. The paper doesn’t show a comparison between the proposed approach and baselines in terms of cost; it would be important to see what loss prediction error is achieved for a certain ratio of small experimental compute cost to large final run cost. We can see this partially in figure 8 which has the largest fitted model parameter size as an x axis. I am convinced that Farseer is better than Chinchilla for relatively large experiment/final cost ratios but there is some hint from this figure that Chinchilla might be better for cheaper predictions which may be more practical for large sweeps of development experiments.

2. It’s also difficult to judge what is a “low enough” loss prediction error. In so far as loss predictions are used to make practical model development decisions it would be important to see if Farseer can make better decisions than Chinchilla on predictive model selection tasks like DataDecide ([https://arxiv.org/abs/2504.11393](https://arxiv.org/abs/2504.11393)).

3. The notation is unnecessarily terse and arbitrary (a whole alphabet of function names). It would be clearer if these single letter function names were expand into short italicized words or to use color and text underneath terms in the equations to clarify what these are. For instance the full schematic of terms “L(N, D) = V (D) + H(N, D) + E + U (N )” ([pdf](zotero://open-pdf/library/items/8LZPGLQW?page=3)) could mark each of the terms (or sets of terms) with the section number in the paper that details them. Likewise explaining “what kind of data is needed for this fitting” up front would be helpful before explaining the whole fitting method. Is this data substantially more expensive to collect than previous scaling laws?

---

> ### Author Rebuttal · Authors · 2025-07-31
>
> Thank you for your detailed, thoughtful, and constructive review. We are encouraged by your positive assessment of our work's significance, originality, and the value of our open-source contributions. We appreciate the specific questions and suggestions for improvement, which will help us strengthen the paper.
> Below, we address the weaknesses and questions you raised.
>
> ---
>
> **W1: Prediction precision and cost.**
>
> You are right to point out that in Figure 8, Farseer's error is comparable to Chinchilla's when fitting on a very small number of models. This is an expected behavior. Because our formula is designed to capture the learning dynamics across a wider range of model sizes (N), it requires more data points across N to realize its full predictive power. When using only a few fitting points from smaller models, its performance naturally converges to the Chinchilla baseline. Its performance is largely comparable to the Chinchilla baseline when using only a few fitting points, and while it shows a very slight disadvantage at the initial point in Figure 8b, its superiority becomes clear as the number and scale of models used for fitting increase.
> Furthermore, a key contribution of our work is the empirical demonstration that Chinchilla's formulation, which lacks the higher-order interaction term H(N, D), leads to divergent prediction errors, as shown in Figure 1a and Figure 2. In contrast, Farseer's error remains convergent and stable across the full range of N.
> The primary reason for this is that the Chinchilla model extrapolates using a single, average rate of data scaling derived from its fitting set to predict performance for all other model sizes. This means that as the target model size N for prediction grows larger and deviates from the fitting range, Chinchilla's error is expected to increase and diverge. Farseer, however, is designed to capture this variance, leading to higher predictive confidence for larger models. This is critical because in most industrial and research applications, the goal of using scaling laws is precisely to forecast the performance of these larger, more resource-intensive models.
> From another perspective, Chinchilla does not inherently benefit from being fitted on a wider range of model sizes (N). Farseer, on the other hand, leverages the additional data points across N to build a more accurate model of how data scaling dynamics change with model size. Therefore, this is not a simple trade-off between prediction accuracy and fitting cost. Instead, it concerns the fundamental capability of the scaling law to generalize reliably to the large-scale regimes that are of greatest practical interest.
>
> **W2: On defining a "low enough" loss prediction error and performance on model selection tasks like DataDecide.**
>
> Our definition of "low enough" is based on the relative error at scale, which approaches the inherent noise from factors like training stability or data parallelism. The key finding is the significant improvement over the Chinchilla baseline within this context. Thank you for the excellent suggestion to evaluate on predictive model selection tasks like DataDecide. This is a great way to measure the practical impact of improved loss prediction, and we will explore incorporating such an evaluation to more directly demonstrate the benefits of our method for development decisions.
> To supplement this, we can share an observation from our industrial practice. When making critical architectural decisions, such as comparing different Attention structures, experiments are typically conducted with models in the scale of tens of billions of parameters trained on hundreds of billions of tokens. In this context, a performance gap between two methods often needs to exceed 0.6% to be considered significant, while the entire performance difference between two competing approaches is rarely more than 3%. However, as shown in Figure 8b, Chinchilla's relative prediction error at the 25.1B scale already exceeds 2.5%. This level of inaccuracy severely undermines its practical value, as the prediction error is nearly as large as the performance margin one hopes to distinguish.
>
> **W3, Q2, Q4, Q8, Q9, Q11, Q15: On notation, clarity, and presentation of the formulas and methodology.**
>
> Thank you for this collection of detailed and very helpful suggestions regarding the clarity of our paper. We sincerely appreciate your careful reading and feedback. We agree that the presentation can be significantly improved. In the revised version, we will perform a thorough update on our notation and presentation based on your feedback. This includes:
> - Adopting more mnemonic function names (e.g., your suggestion of L_N(N) instead of U(N)).
> - Adding direct, plain-language explanations of terms like A(N), B(N), G(.), and U(.) to the figure captions to make them self-contained.
> - Moving the overview of experiments (Figure 7) earlier to clarify the data requirements for our fitting method from the outset.
> - Explicitly stating that all parameters other than N and D are fitted.
> - Restructuring the beginning of Section 3.2 to more clearly state the motivation and logic of our fitting procedure.
> We believe these changes will substantially improve the paper's readability and logical flow.
>
> **Q1 & Q3: On the release of training code and the curated validation set.**
>
> Thank you for your interest. We are open-sourcing the Farseer fitting code. The training code, which is a modified version of the open-source project Megatron-LM, will not be released. Our primary modifications involve custom implementations to accelerate Data Parallelism (DP) and Tensor Parallelism (TP). For the smaller models in our study, which do not require complex parallel strategies, we believe that training with the standard community version of the codebase would not significantly impact the results.
>
> Furthermore, ensuring the validity and unbiased nature of the validation set was a resource-intensive endeavor, requiring substantial manual effort to maintain impartiality in the data curation process. For instance, the book portion of the set was fully annotated by human reviewers, rather than relying solely on machine-based OCR. We understand the value that releasing this validation set would provide to the community for reproducing our work. Consequently, we are actively seeking approval to make this dataset publicly available in the future.
>
> The 2024 data cutoff for the English validation set is indeed intended to ensure it contains data unseen by the models, as its academic paper sources are from arXiv publications after 2024. We can confirm that we constructed similar validation sets for Chinese and code, and the fitting performance was also strong on these, though the English validation set was the primary one used for the results reported in the paper.
>
> **Q5: Regarding the specific slice shown in Figure 2.**
>
> In Figure 2, the model size is fixed at N=6B. We chose this specific slice to provide a clear, direct comparison between the Farseer and Chinchilla functional forms against the ground-truth loss points. This visualization effectively highlights the improved fit our proposed form provides for a given model size across varying data sizes.
>
> **Q6 & Q7: On the role of scaling laws and the meaning of the "scaling gap".**
>
> To clarify our statement that LM progress is "attributed to scaling laws," we mean that the discovery and use of these laws have given researchers the confidence to invest massive compute resources, knowing they can be predictably converted into model capability. Scaling laws have been essential for guiding and evaluating this process.
> By "scaling gap," we refer to the phenomenon where insights and conclusions drawn from small-scale experiments (e.g., identifying a superior dataset or config) do not consistently hold true when scaling up to much larger models. A more accurate scaling law, like Farseer, can help mitigate this by providing more reliable predictions of large-scale performance, thereby making small-scale exploratory experiments more predictive and valuable. In fact, Appendix E provides a simple example from this scaling-law perspective.
>
> **Q10: Regarding the x-axis in Figure 6 and its relation to Figure 8.**
>
> You are correct. The x-axis label in both Figure 6 and Figure 8 is "Largest model size used in fitting". The figures do show similar trends but are complementary as they show extrapolation to different target model sizes. We agree Figure 8 is more informative with the Chinchilla baseline and will consider merging or refining the presentation to avoid redundancy.
>
> **Q12, Q13, Q14, Q16: On various typos and clarifications.**
>
> Thank you for spotting these. You are correct on all points.
> - Line 86: The reference should indeed point to Appendix A, which details hyperparameter and architecture ablations. Appendix F contains the specific model settings.
> - Figure 3: We will increase the tick density on the axes. The y-axis direction indicating faster loss improvement will be clarified. The R² value shown is the average R² across each of the fitted lines. We will clarify the caption and define terms before they are used.
> - Line 130: We will correct the LaTeX typo.
> - Line 162: You are right, the reference should be to Figure 5(b).
>
> We will correct all of these in the revision. Thank you again for your meticulous review.

---

> > ### Comment · Reviewer_B5rQ · 2025-08-06
> > **Thank you for response**
> >
> > I appreciate the authors detailed response. My concerns previously were primarily about polishing the clarity of presentation for what is already a good paper, and from the authors response I can see they have a good plan for revision. I will increase my score.

---

### Official Review · Reviewer_XmBY · 2025-07-02

**Clarity:** 4
**Significance:** 4
**Originality:** 3
**Rating:** 5
**Confidence:** 4

**Summary:**

This paper introduces Farseer, a novel scaling law for large language models (LLMs) that significantly improves prediction accuracy by refining the modeling of the loss surface  L(N, D). The core innovation of Farseer lies in its unique coupling mechanism between model size  N and data size D , which demonstrates a clear advantage over the fixed parameter assumption of the Chinchilla scaling law. The research team validated the effectiveness of this method through large-scale experiments on approximately 1,000 LLMs, consuming 3 million H100 GPU hours. The results show that Farseer achieves an error of less than 0.5% when extrapolating large-scale performance from small-scale experiments. Furthermore, the study reveals the variation in the optimal D/N ratio as the computational budget increases, providing crucial guidance for resource allocation in practical training scenarios.

**Questions:**

1. The loss surface modeling of Farseer adopts an exponential nesting form, but the paper does not theoretically explain why this form is most suitable for describing the scaling behavior of LLMs. Are there other theoretical bases supporting this design? It is recommended to supplement the theoretical analysis, for example: through the convergence analysis of gradient descent, illustrate how the exponential terms reflect the variation in optimization difficulty with scale; or compare with other functional forms (e.g., polynomial, logarithmic combinations) to prove the necessity of the current design.
2. The largest model validated in the experiments has 25.1B parameters, but the current mainstream LLMs have reached the trillion scale. Will the prediction error of Farseer increase under ultra-large scales?
3. The experiments are based solely on the Transformer architecture and bilingual (Chinese-English) data, but the LLM field has seen the emergence of new architectures such as MoE and multimodal systems. It is recommended to supplement the experiments across architectures or data types.

**Ethical Concerns:**

["NO or VERY MINOR ethics concerns only"]

**Final Justification:**

Thanks for the solid work and nice responses by the authors. I have carefully read the authors' responses to my concerns and comments, as well as the interactions between the authors and all reviewers. Almost all reviewers' concerns have been satisfactorily resolved. Therefore, in my opinion, it meets the standards required by the conference.

**Limitations:**

The authors clarified the primary reason for limiting the validation to LLMs with a maximum of 25B parameters and not including experiments with larger models. They outlined follow-up research plans focusing on MoE architectures and code data. The authors also explained the rationale for choosing BPC instead of downstream task evaluations. They acknowledged the common issue of lacking theoretical explanations in the scaling law domain. However, while mentioning that large-scale training requires complex parallelization strategies that introduce performance noise, they did not quantify the impact of this noise on Farseer's prediction error.

**Paper Formatting Concerns:**

No major formatting issues were identified.

**Quality:**

4

**Strengths And Weaknesses:**

Advantages:
1.Highly innovative: Farseer scaling law explicitly couples model size (N) and data size (D), significantly improving prediction accuracy compared to existing methods such as Chinchilla.
2.Extensive experimental validation: Based on experiments with 1,000 LLMs of varying scales (3 million H100 GPU hours), the conclusions are highly credible due to the abundance of data. When extrapolating large-scale performance from small-scale experiments, the prediction error is less than 0.5%, which is markedly superior to benchmark methods like Chinchilla.
3.Rigorous methodology: Employing differential analysis and multi-stage fitting strategies avoids overfitting and ensures model generalizability.
4.Outstanding practical value: Provides quantifiable resource optimization schemes for LLM training, reducing trial-and-error costs.
Disadvantages:
1.The formula design relies on empirical fitting and lacks a solid theoretical explanation.
2.The maximum validated model size is only 25.1B parameters, without covering the scale of hundreds of billions, raising concerns about the reliability of extrapolation to ultra-large scales. Additionally, the research is limited by computational capacity, with validation restricted to Llama-style decoder Transformers, failing to encompass a broader range of model architectures.
I look forward to further research based on Farseer!

---

> ### Author Rebuttal · Authors · 2025-07-31
>
> We sincerely thank you for your thorough review and positive evaluation of our work. We appreciate your recognition of Farseer's innovation, extensive validation, and practical value. Your questions are insightful, and we would like to address them below.
>
> ---
> **Disadvantage 1 & Q1: Regarding the theoretical explanation for the formula's exponential form**
>
> You correctly pointed out that our paper lacks a deep theoretical explanation for the chosen exponential form. We agree that a complete theoretical justification for a scaling law's functional form is a challenging but important goal. While a first-principles derivation remains an open problem, we did empirically validate our choice. To help the community tackle this open problem, we have open-sourced our extensive experimental data, models, codes and training curves.
> As you suggested, we compared our exponential form against other functional forms. Due to space constraints, these results were included in Appendix C.3, Table 2. The results clearly demonstrate that our proposed formula provides the best fit among the alternatives we tested, justifying our choice from an empirical standpoint.
>
> **Disadvantage 2 & Q2: Regarding the maximum validation scale of 25.1B parameters**
>
> You raised a valid concern about the extrapolation of Farseer to models larger than our maximum validated size of 25.1B parameters. We acknowledge this limitation. This was primarily due to constraints on computational resources and the increasing engineering complexity of stable, large-scale parallel training. We believe our experiments up to 25.1B provide strong evidence for Farseer's effectiveness, and we acknowledge that validating on even larger models is a critical next step as resources permit.
>
> **Q3: Regarding the scope of model architectures and data types**
>
> Thank you for this excellent suggestion. We agree that testing across diverse architectures and data modalities is crucial for generalizability. We are pleased to clarify two points:
> - Data Types: In Section 4.1 and Figure 6, we demonstrate Farseer's robustness across different data distributions. Furthermore, as part of our ongoing work, we have conducted a detailed study of the code domain, including the effects of various code-mixing ratios. These experiments confirm that the Farseer scaling law continues to yield highly accurate predictions. We believe a full analysis of these results is an important direction for future work and plan to report them separately.
> - Architectures: As for architectures like Mixture-of-Experts (MoE), they introduce additional complexities and variables (e.g., sparsity, fine-grained routing) that require more detailed modeling and experimentation. This is an exciting direction that we have planned for our future work.
>
> We hope these clarifications address your concerns and further highlight the contributions of our work. Thank you once again for your valuable feedback.

---

> > ### Comment · Reviewer_XmBY · 2025-08-07
> >
> > Thanks for your response, which has addressed my concerns. I will keep my rating.

---

### Note · Authors · 2025-08-16

Dear Area Chair and Reviewers,

We are deeply grateful for the thoughtful and constructive reviews, which have significantly strengthened our paper. We are pleased that the rebuttal discussion successfully resolved all initial concerns, culminating in a strong positive consensus. We are especially encouraged that Reviewers #B5rQ and #NtZX kindly confirmed they would raise their scores.

We are greatly encouraged that all reviewers recognized Farseer as a highly innovative scaling law. It establishes a new fundamental paradigm that solves the divergent extrapolation errors fundamentally limiting prior laws by effectively modeling the interaction between model and data size (Reviewer #XmBY, #B5rQ, #gWVt, #NtZX).

We are also pleased that reviewers recognized the rigorous and extensive empirical validation of our work, noting its outstanding practical value and Farseer's demonstrated robustness and generalizability. (Reviewers XmBY, B5rQ, gWVt).

Finally, we are grateful that the paper was found to be clearly written and well-structured, and that our comprehensive open-source release was highlighted as a valuable contribution to the community (Reviewer #B5rQ, #gWVt).

We sincerely thank all the reviewers for their invaluable guidance. Their suggestions have made our work stronger, and we will diligently incorporate all feedback into the camera-ready version.


Sincerely,

The Authors of Submission 22519

---

### Decision · Program_Chairs · 2025-09-17

**Decision:**

Accept (spotlight)

**Comment:**

This paper proposes Farseer, a refined scaling law for large language models that explicitly models the interaction between model and data size. The work is supported by an unusually large empirical study (~1,000 trained models) and a comprehensive open-source release.

Reviewers agreed that the contribution is both novel and practically important, showing clear improvements over the widely used Chinchilla law in terms of predictive accuracy and stability. The main concerns were the lack of a strong theoretical foundation for the functional form, the limited maximum scale of validation (25B parameters), and some clarity issues in notation and presentation. The rebuttal addressed these issues satisfactorily, clarifying empirical justification for the functional form, explaining resource constraints, and committing to substantial improvements in clarity and limitations discussion.

Following the rebuttal, reviewer consensus was strongly positive, with multiple score increases. Overall, this is a technically solid and impactful paper that is likely to influence future research and practice in scaling laws, in particular, through its open source release of data.